# ARE SYNTHETIC CLASSIFIERS REALLY AS GOOD AS REAL CLASSIFIERS?

## ABSTRACT

Foundation models have achieved significant advancements across various domains, yet their training demands vast amounts of real-world data, which is becoming increasingly scarce. To address this challenge, synthetic data has garnered substantial interest as an alternative for augmenting training datasets in fields such as computer vision and natural language processing. However, skepticism remains regarding whether synthetic classifiers can match the performance of those trained on real data. In this paper, we investigate this question by conducting a detailed analysis within the realm of visual tasks, comparing classifiers trained on synthetic versus real data using CLIP and ViT. Our results reveal that synthetic classifiers exhibit deficiencies in a range of challenging real-world scenarios, such as fine-grained classification, extreme object scales and extreme brightness despite achieving comparable overall accuracy to their real-data-trained counterparts. We find that the limitations of synthetic classifiers can be traced back to the limitations of current generative models in capturing the complexity and diversity of real-world data in these aspects. To mitigate these issues efficiently, we explore **RealTune**, a simple method that enhances synthetic classifiers by finetuning them with a small amount of real data. Experimental evaluations demonstrate that RealTune significantly improves the performance of synthetic classifiers using only a limited real dataset (*e.g.,* 40k images, 3% of ImageNet) with minimal training time (*e.g.,* 1hour on a single NVIDIA RTX 3090 GPU). Our findings indicate that while synthetic data is a valuable resource, integrating real and synthetic data is essential to achieve robust and efficient classifiers. This work underscores the necessity of leveraging both data types to bridge the performance gap and enhance the overall effectiveness of foundation models.

## 1 INTRODUCTION

Despite remarkable advancements across various fields, foundation models necessitate vast amounts of training data (Brown et al., 2020; Radford et al., 2021), posing challenges as the availability of real-world data becomes increasingly limited (Villalobos et al., 2024). As a result, synthetic data has garnered significant attention as an alternative for generating training data across different domains (Sankaranarayanan et al., 2018; Hwang et al., 2024; Kollias, 2022; He et al., 2022). Although there are widespread concerns that synthetic data may contaminate and degrade model performance (Hataya et al., 2023; Shumailov et al., 2024; Dohmatob et al., 2024b;a), recent studies provide promising evidence that synthetic classifiers trained solely on synthetic data can achieve performance comparable to real classifiers in ImageNet classification (Tian et al., 2023; Fan et al., 2024; Tian et al., 2024).

However, the current debate primarily centers on comparing the *learning outcomes* (*e.g.,* ImageNet accuracy), overlooking the detailed connections between these outcomes and the training data—a relationship crucial for future model designs. To foster a more constructive discussion, this work presents a *fine-grained, quantitative* analysis of the real-world behaviors of synthetic classifiers and traces these distinctive behaviors back to their origins in the training data. This approach enhances our understanding of the gap between real and synthetic data in model training.

Specifically, we focus on vision tasks as a case study, examining two classes of visual foundation models: the supervised classifier ViT (Dosovitskiy et al., 2021) and the visual-language model

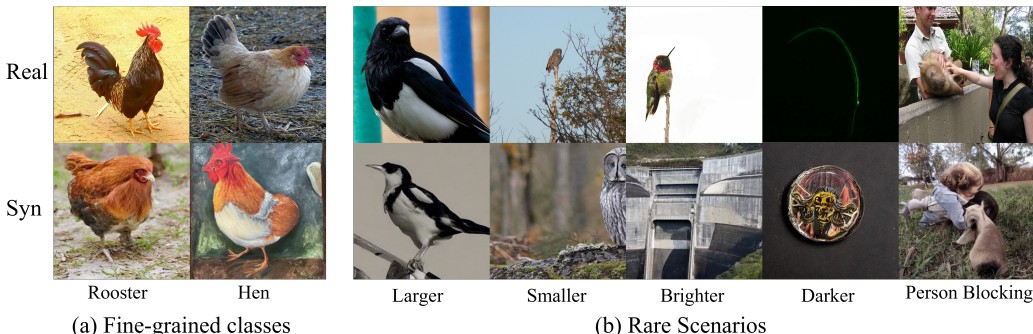

Real
Syn

Rooster       Hen          Larger       Smaller     Brighter     Darker    Person Blocking

(a) Fine-grained classes                  (b) Rare Scenarios

Figure 1: Illustrative comparison of real and synthetic data across multiple challenging scenarios. (a): Illustrating semantic confusion in fine-grained classes within synthetic data, the synthetic rooster lacks a comb while the synthetic hen possesses them, contrary to the real distinction where roosters are identified by their combs, a feature absent in hens. (b) In rare scenarios, synthetic images underperform compared to real data, exhibiting limited diversity in scale variations of central objects, lacking brightness variations, and struggling to effectively generate images blocked by a person.

CLIP (Radford et al., 2021), where real and synthetic data are shown to have similar overall performance (Fan et al., 2024). However, through comprehensive evaluation, we identify several scenarios where synthetic classifiers struggle, including: 1) similar images with *fine-grained differences* (*e.g.,* rooster and hen), 2) rare images exhibiting *unusual object scales and brightness*, and 3) complex situations involving *person blocking*. These findings suggest that synthetic classifiers, despite achieving comparable benchmark accuracies, may still underperform in challenging real-world scenarios.

But how do these deficiencies arise? We conduct a detailed quantitative study to trace their origins in the training data. Specifically, we quantify the gap between real and synthetic data by developing a suite of measures for: 1) fine-grained semantic consistency, 2) object scales and brightness, and 3) detection of person blocking. Our findings reveal that, although synthetic images often appear realistic to the human eye, *at a distributional level*, **current generative models still struggle to achieve the same level of accuracy** in representing fine-grained semantics, **diversity** in object scales and brightness, and **complexity** in scenarios like person blocking as compared to real data; see illustrations in Figure 1. Further controlled studies on three core elements of synthetic data generation—generative models, text prompts, and classifier guidance—indicate that, while these elements provide some assistance, we are currently unable to bridge these fundamental gaps between real and synthetic data in these challenging scenarios.

Finally, rather than attempting to bridge this gap by increasing computational resources, we propose a more efficient and effective approach, **RealTune**, which is to simply finetune synthetic classifiers using a small amount of real data. We demonstrate that RealTune not only significantly improves overall accuracy but also rapidly mitigates the identified gaps in challenging scenarios. Our ablation study reveals that RealTune is considerably more efficient than alternative methods, such as finetuning real classifiers with synthetic data. Moreover, combining RealTune with mixed pretraining on both real and synthetic data—a strategy suggested by Wang et al. (2024)—enables classifiers to outperform both real and synthetic counterparts by a substantial margin (up to 17.2% on ImageNet-100). To summarize, our contributions are:

- We pinpoint key challenging scenarios for synthetic classifiers, including fine-grained image distinctions, unusual object scales and brightness, and complex person blocking.

- We conduct a fine-grained study of these scenarios through a suite of quantitative measures, and demonstrate fundamental discrepancies between real and synthetic training data in fine-grained semantic consistency, diversity, and complexity.

- We investigate **RealTune**, an efficient method to bridge these gaps by finetuning synthetic classifiers using a minimal amount of real data, showing that a mixture of real and synthetic data can combine the best of both worlds.

## 2 A FINE-GRAINED ANALYSIS OF REAL AND SYNTHETIC CLASSIFIERS

In this section, we conduct a detailed comparison between real and synthetic classifiers, characterizing several key differences when deploying them to real-world scenarios.

Table 1: Overview of real and synthetic classifiers in our analysis.

| Model | ViT | | | | CLIP | | |
|---|---|---|---|---|---|---|---|
| Training Data | ImageNet | | Synthetic | | LAION | | Synthetic |
| Data Size | 0.25M | 1M | 1M | 2M | 64M | 371M | 371M |
| ImageNet Acc | 58.21 | 78.64 | 52.51 | 58.72 | 55.12 | 66.77 | 55.68 |

**Experiment Setup.** Following SynRep (Fan et al., 2024), we consider two commonly used types of visual backbones: supervised ViT (Dosovitskiy et al., 2021) and the vision-language model CLIP (Radford et al., 2021) (both using ViT-Base (ViT-B) backbone), pretrained on different sources and sizes of training data.[1] We list the model statistics in Table 1. To facilitate discussions, we use notations like CLIP-Real-64M (a CLIP model trained on 64M real data). For a fair comparison between real and synthetic classifiers, we consider two settings: 1) **equal data size**, where two classifiers are obtained from the same amout of data, such as CLIP-Real-371M and CLIP-Syn-371M, ViT-Real-1M and ViT-Syn-1M; 2) **equal accuracy**, where the two models have close test accuracy on ImageNet, such as CLIP-Real-64M and CLIP-Syn-371M, ViT-Real-0.25M and ViT-Syn-2M.

### 2.1 QUANTITATIVE COMPARISON IN CHALLENGING SCENARIOS

To obtain an evaluation beyond standard benchmarks, we begin by comparing the performance of real and synthetic classifiers in challenging scenarios: fine-grained classification and rare scenarios.

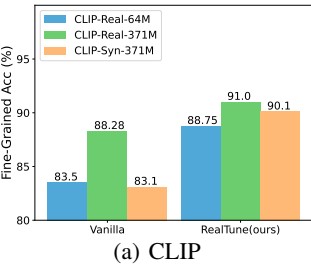 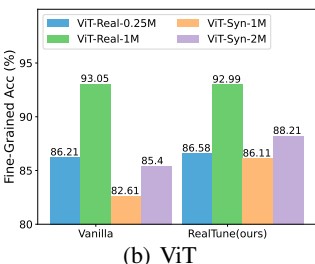

(a) CLIP  (b) ViT

Figure 2: Comparing real and synthetic classifiers at fine-grained classification. We conduct the experiment on ImageNet and calculate the fine-grained accuracy within each coarse-grained class by constraining the label spaces accordingly.

**Synthetic Classifiers Struggle with Fine-grained Classification.** Real-world images contain concepts in different granularities. For example, a coarse-grained class "dog" contains over 100 dog species in ImageNet (*e.g.,* golden retriever), *i.e.,* a variety of fine-grained classes. We hypothesize that since generative models are often worse at following fine-grained instructions during generation (Saharia et al., 2022), they may suffer at fine-grained classification. Leveraging the hierarchical labels in ImageNet, we calculate the fine-grained accuracy for discriminating classes *within* each coarse-grained class, by constraining the label spaces accordingly. Figure 2 shows that real classifiers have much better fine-grained accuracy, especially when pretrained on the same data size. Even comparing models with similar overall accuracy (CLIP-Real-64M and CLIP-Syn-371M, ViT-Real-0.25M and ViT-Syn-2M), real classifiers still attain better performance at fine-grained classification [2]. It shows that synthetic classifiers particularly struggle at discriminating fine-grained classes.

---

[1]We directly adopt the checkpoints provided by the official SynRep repository for reproducibility.

[2]ImageNet accuracy's variation is usually at most 0.3 and in real world, the variance/stdev is usually much smaller. For example, Dosovitskiy et al. (2021) report 85.30 ± 0.02 for ViT and 87.54 ± 0.02 for ResNet (Table 2 in Dosovitskiy et al. (2021)). Therefore, the fine-grained accuracy difference of models at equal performance is a clear difference.

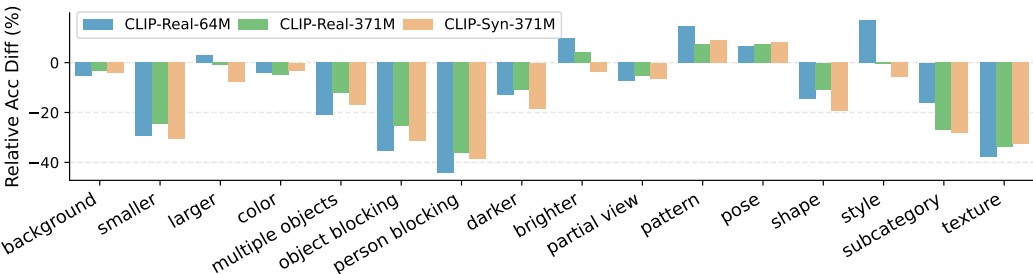

Figure 3: Rare scenario robustness of CLIP on ImageNet-X. Higher is better. ViT results are shown in Figure 8 in Appendix.

**Synthetic Classifiers Struggle with Rare Scenarios.** Real-world visual applications often contain rare scenarios that are observed less often during training. To compare the ability to generalize to rare scenarios, we evaluate real and synthetic classifiers on ImageNet-X (Idrissi et al., 2023), benchmarking the robustness of image classification *w.r.t.* 16 rare scenarios, such as background, texture, object scale, object blocking, brightness, *etc*. For each scenario, we calculate the relative difference in accuracy, $(\mathrm{acc_{rare}} - \mathrm{acc_{all}})/\mathrm{acc_{all}}$, as a measure of the influence ratio, where $\mathrm{acc_{all}}, \mathrm{acc_{rare}}$ refer to the accuracy on ImageNet validation set and a specific rare scenario of ImageNet-X, respectively.

The evaluation results for CLIP are shown in Figure 3. First, CLIP-Syn-371M underperforms CLIP-Real-371M in 12 out of 16 scenarios, indicating a noticeable distinction between the synthetic and real classifier. Second, for some scenarios including **multiple objects, object blocking, and person blocking**, CLIP-Syn-371M underperforms CLIP-Real-371M (equal data size) while outperforming CLIP-Real-64M (equal accuracy). This indicates that the synthetic dataset contains data with corresponding scenario but this is still relatively *scarce* compared to the real dataset. Finally, CLIP-Syn-371M performs worse at some scenarios such as **smaller, larger, darker and brighter** compared to real CLIP under equal data size and equal accuracy. Specifically, it indicates that **synthetic classifiers fundamentally struggle at processing extreme object scales and image brightness**. Additionally, Singh et al. (2024) also arrived at a similar conclusion that the performance of synthetic classifiers on ImageNet-C (Hendrycks and Dietterich, 2018) and ImageNet-3DCC (Kar et al., 2022) is significantly lower than that of real classifiers. We hypothesize that this is caused by a lack of variation in diffusion-generated images, a question we will explore in Section 3. Similar conclusions hold for ViT results (see Figure 8 in Appendix) .

> **Takeaways of Section 2**
>
> We identify several key limitations of synthetic classifiers in real-world applications:
> - Synthetic classifiers struggle to **discriminate fine-grained classes with similar semantics**.
> - Synthetic classifiers struggle with **rare scenarios *w.r.t.* object sizes, brightness and complex scenes such as person blocking.**

# 3 DEVIL IN THE DATA: QUANTITATIVE EXAMINATION OF SYNTHETIC DATA

In Section 2 we observed that despite having similar performance on certain benchmarks, synthetic classifiers struggle in many real-world scenarios. Since the only difference between real and synthetic classifiers is training data, the data quality is the key to understanding this gap. Hence, next, we examine the disparity between real and synthetic training data. For an initial qualitative understanding, we illustrate some manually picked real and synthetic examples in Figure 1. More rigorously, we develop quantitative measures of data quality for each scenario, which we collectively denote as **SynBench**. These metrics can be used for benchmarking the progress of synthetic data on these aspects, which may be of independent interest.

**Setup.** Given that ImageNet contains a large number of classes to be generated, for better efficiency, we conduct experiments on ImageNet-100, a 100 class subset of ImageNet that is commonly used in

visual tasks (Tian et al., 2020). We randomly select 50 images from each class (5k images in total) as the real dataset. For a consistent setup, we strictly follow the default settings in SynRep (Fan et al., 2024) for generating equivalent ImageNet-like images as the default synthetic dataset. Specially, the default synthetic images are generated using Stable Diffusion V1.5 (SD-V1.5) with a classifier-free guidance (CFG) scale of $\omega = 2$ and IN-Caption format prompts (class name + captions, generated by BLIP2 (Li et al., 2023), *e.g.,* "Tench, a man holding a fish"), which is the optimal configuration for generating synthetic data for the synthetic ViT in SynRep. We will consider three main aspects in image generation:

**F1**: **Classifier-free guidance.** Classifier-free guidance (CFG) (Ho and Salimans, 2021) is a common technique to align image generation with the prompt. A large CFG scale $\omega$ ensures better alignment but sacrifices the diversity of synthetic images. We explore changing CFG from 2 to 7 to investigate the impact of different CFG scales on synthetic data.

**F2**: **Text prompts.** In text-to-image models, text prompts determine the main semantics of synthetic images. The default IN-Caption format prompts may lack sufficient variation. We explore adding phrases describing the specific rare scenarios (*e.g.,* in a dark environment). See Appendix A.1 for more details.

**F3**: **Generative models.** Different generative models have different capacities depending on their model size and training methods. Apart from SD-V1.5, we include three other models: 1) Stable Diffusion V2 (SD-V2) (Rombach et al., 2022) that enhances the text-encoding capacity and the diversity of training data compared to SD-V1.5; 2) Sing Diffusion (Zhang et al., 2024) modifies the sampling method of SD to tackle brightness issues; 3) DeepFloyd IF (Shonenkov et al., 2023) is a generative model distinct from SD, exhibiting a high degree of photorealism and language understanding. Figure 10 in Appendix provides examples generated by these models.

### 3.1 SYNTHETIC DATA HAVE HIGH FINE-GRAINED CLASS CONFUSION

**Fine-grained Class Confusion.** As discussed in Section 2.1, we find that synthetic classifiers struggle with discriminating similar but different fine-grained classes, *e.g.,* roosters and hens (both are chicken). We find that the essential cause lies in a problem that we refer to as *Fine-Grained Class Confusion*, where generative models cannot faithfully follow instructions and generate the corresponding fine-grained classes. As illustrated in Figure 1 (a), the generative models confuse roosters and hens, even when being explicitly instructed.

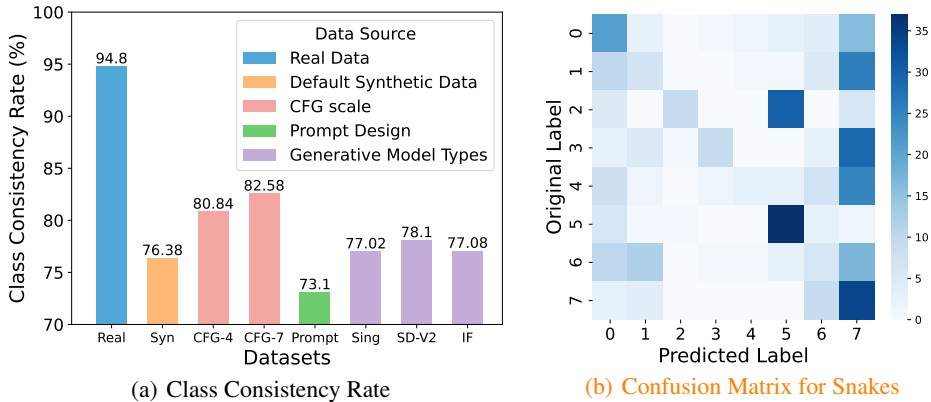

(a) Class Consistency Rate  (b) Confusion Matrix for Snakes

Figure 4: Quantitative measures of class confusions in real and synthetic data. (a): Class consistency rate for real (blue bars) and synthetic data of different types. Orange bars represent default synthetic data, red bars show CFG scale adjustments, green bars indicate prompt format changes, and purple bars reflect generation model type changes. (b): Fine-grained confusion matrix between original label (used in the prompt to generated the image) and predicted label (predicted by ConvNeXt-B) of the 8 snake species in synthetic ImageNet-100.

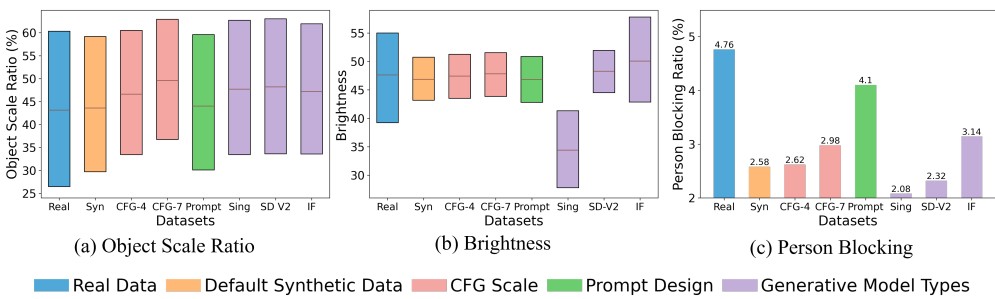

Figure 5: (a): Box plots illustrating the distribution (25%, 50%, and 75% quantiles) of object scale ratio in different datasets. (b): Box plots illustrating the distribution of brightness. (c): The proportion of images with person blocking.

**Measurements.** Quantitatively, we measure fine-grained class confusion by the *consistency rate* between the original labels[3] and the predicted labels with a state-of-the-art classifier ConvNeXt-B (Liu et al., 2022) with 91.20% top-1 accuracy on ImageNet-100. A low consistency rate indicates that the actual image semantics are inconsistent with the original labels (or the image semantics are hard to distinguish). Figure 4(a) shows that while real data have a high consistency rate, synthetic data have a surprisingly low consistency rate of 76.38%, indicating that *about a quarter of synthetic images have wrong fine-grained labels*. Although this score may be affected by the chosen classifier, this sharp contrast in class consistency still strongly indicates that fine-grained class confusion is a common issue in synthetic data and these inaccurate pairing between images and labels (descriptions) can hamper their performance of the trained synthetic classifier. Figure 4(b) illustrates the confusion matrix of 8 snake species in synthetic data as an example for a more intuitive understanding. It shows that the 8 snake species are predominantly predicted as class 5 (vine snake) and 7 (horned viper), intuitively indicating the presence of fine-grained class confusion issues in the synthetic data.

**Mitigating Class Confusion.** At last, we explore whether refine prompts, CFG scale and alternative generative models can resolve this issue. For prompts, we try to include only the fine-grained class names {class_name} to avoid other descriptions in the ImageNet caption that might distort semantics unexpectedly. As shown in Figure 4, we find that adjusting CFG scale to be larger can increase class consistency but is still far from closing this gap; while editing prompts and using different generative models lead to little improvement. Thus, we conclude that fine-grained classes are much harder for generative models today to distinguish (may require even larger model sizes and compute), while real data still have significant advantages.

## 3.2 THE LACK OF RARE SCENARIOS IN SYNTHETIC DATA

Next, we study the reason for the inability of synthetic classifiers to distinguish some rare scenarios, in particular, object scale, brightness, person blocking (see Figure 3). Similarly, we find that although it is very easy to collect extreme samples in the real world (*e.g.,* large and small objects), they are often hard to synthesize in existing generative models, as illustrated in Figure 1. Below, we design quantitative measures for each rare scenario, and examine whether adjusting prompts, CFG scale and generative models could alleviate these obstacles.

**Object scale.** Object scale (larger and smaller) influences the proportion of the central object within the entire image. Quantitatively, we use YOLO world (Cheng et al., 2024), an open world object detection model, to find the central object and use the proportion of the area occupied by the central object in the image, as a measure of the object scale. The results are shown in Figure 5(a). We can see that synthetic data (orange column) indeed have a smaller range of object scales compared to real data, showing that synthetic data lack very small and very large objects. Adjusting prompts (to explicitly include "large" and "small" keywords) hardly improves the range. Increasing CFG or using other models will introduce a *systematic shift* in scale range, mostly toward larger scales. It suggests that the limitation of generating large objects can be addressed by adjusting CFG scales but it remains hard to generate small objects.

---

[3]Here the original label is used in the prompt to generate the input image, which can be mismatched to the actual semantics of the synthetic image due to the imperfection of underlying generative models.

**Brightness.** Next, we look at the overall brightness of the image, where we measure brightness in the CIELAB color space that is known to be more perceptually aligned (Wyszecki and Stiles, 2000). As depicted in Figure 5(b), real images have a much wider range of brightness than the default synthetic data. Modifying the prompt (by adding phrases like "in a dark environment" or "in a bright environment" to the default prompt) or increasing CFG leads to limited improvements. Sing Diffusion (Zhang et al., 2024), a model specifically designed to tackle brightness issues in Stable Diffusion, yields a systematic shift towards darker images, often generating disruptions such as a predominantly black background or entirely black images as shown in Figure 10. On the other hand, DeepFloyd IF (Shonenkov et al., 2023) can generate brighter images but falls short in producing darker samples. In summary, existing models can hardly achieve a proper and diverse brightness range like real images.

**Person blocking.** Blocking indicates whether the central object is blocked by a person. We first use YOLO-V5 (Ultralytics, 2021) to select images containing people. Then we use GPT-4o (Achiam et al., 2023) to identify whether the central object in these images is blocked by a person by asking, "Is part of {class_name} occluded by the human body in the image? Please only answer yes or no." Figure 5(c) shows that 4.76% of images in real data are person blocked, while this ratio is only 2.56% in synthetic images (similar for other generative models). We find that explicitly instructing the model to generate person-blocking images (adding "occluded by human body" to prompts) can signfiicantly alleviate this issue, though not enough to close the gap. Besides, we observe that the generated person-blocking images are often distorted (see examples in Figure 1), indicating that it is hard for existing generative models to generate realistic complex scenes like person blocking.

> **Takeaways of Section 3**
>
> The ineffectiveness of synthetic classifiers stems from the inability of current generative models to generate **faithful fine-grained semantics**, **diverse object scales**, **high-range brightness**, and **complex scenes**. And these limitations *cannot* be easily remedied by adjusting prompts, CFG scale or generative models.

## 4 THE IMPACT OF REAL DATA ON SYNTHETIC CLASSIFIERS

Section 3 shows that even if prominent generative models like Stable Diffusion are able to generate very realistic-looking examples, from a *distributional* perspective, the synthetic data are still strongly biased, and thus have a significant gap to real data when used for model training.

According to the scaling laws of text-to-image models (Li et al., 2024), resolving these problems with stronger generative models would cost much more data, much larger networks, and much more computing (for both training and inference), which significantly increases energy consumption and carbon footage. Instead, as we show in Section 3, **randomly sampled real data**, which do not have these problems, easily beat synthetic data in many challenging aspects. In this sense, real data can be a critical lever for us to resolve the limit of synthetic data in a *dramatically more efficient way*.

Motivated by this observation, we examine the impact of real data on synthetic classifiers under a simple strategy, that is to finetune the pretrained synthetic classifier with *a small amount of randomly sampled real data*. We call it **RealTune**. Compared to conventional paradigms that directly pretrain with large-scale real data (which may be unsubstantial in the future), we advocate for pretraining with large-scale synthetic data (which is easier to *reproduce*) and remedying its defects by finetuning with a small amount of real data.

**Setup.** We conduct our finetuning experiments using the pretrained real and synthetic ViT and CLIP models. For each classifier, we consider three settings: 1) **Vanilla** with no finetuning (baseline); 2) **SynTune**, which finetunes models on synthetic data generated by Stable Diffusion following the setup in Section 3; and 3) our **RealTune**, which finetunes models on real data randomly sampled from the ImageNet training set. The finetuning data comprises 40k samples (which is only 3% of ImageNet), with finetuning conducted for 50 epochs for CLIP and 30 epochs for ViT. We evaluate the resulting model on the ImageNet validation set. See Appendix A.2 for more experimental details. Results are summarized in Figure 6. Additionally, we also conducte experiments by finetuning models on ImageNet-V2 (Recht et al., 2019) and evaluating on ImageNet to circumvent the benefits of in-domain data finetuning. The results can be found in Figure 9 in Appendix, and the conclusion is consistent with finetuning on in-domain data.

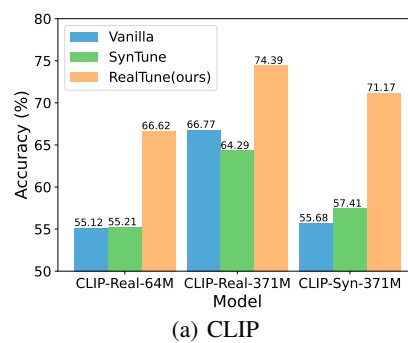 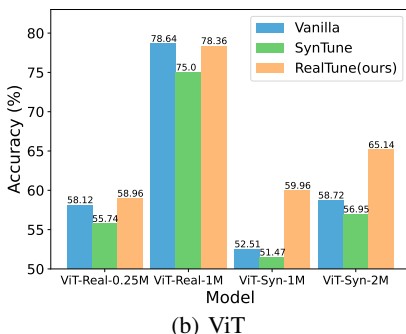

(a) CLIP · (b) ViT

Figure 6: ImageNet classification accuracy of different finetune methods.

## 4.1 RealTune Bridges the Performance Gap Efficiently

**RealTune significantly improves the accuracy of synthetic classifiers.** Specifically, RealTune achieves an improvement of 7.45% on ViT-Syn-1M, 6.42% on ViT-Syn-2M, and 15.49% on CLIP-Syn-371M by using only a small amount of real data and a short training duration as shown in Figure 6. In contrast, the accuracy of real ViT declines after finetuning with real data, indicating that real data are particularly helpful for synthetic data while being non-helpful for real classifiers (even leading to overfitting and degradation). Likewise, synthetic data are not helpful for synthetic classifiers, either. Thus, the only gain from cross-source finetuning is RealTune, because real data can remedy the limitations of synthetic data.

**With RealTune, synthetic classifiers outperform real classifiers when pretraining accuracy is comparable.** Specifically, CLIP-Syn-371M surpasses CLIP-Real-64M by 4.55%, and ViT-Syn-2M surpasses ViT-Real-0.25M by 6.18% as shown in Figure 6. Moreover, RealTune decreases the accuracy gap between real and synthetic classifiers significantly at equal pretraining data size. Notably, the gap for CLIP decreased from 11.09% to 3.22% . From this perspective, RealTune can mitigate the requirement for extensive real data in CLIP training (Radford et al., 2021).

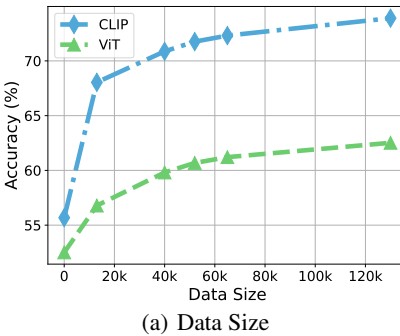 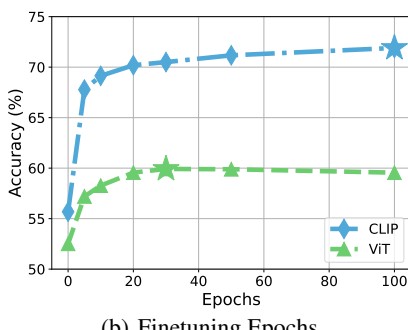

(a) Data Size · (b) Finetuning Epochs

Figure 7: Ablation study on (a): training data size; and (b): total finetuning epochs. Both plots start with the default pretrained model (when data size or epoch equals to zero).

At last, we investigate the impact of two factors of RealTune: data size and finetuning epochs. Figure 7(a) illustrates that using a small size of real data (40k, 3% of the ImageNet trainset) for RealTune rapidly enhances model performance, with better outcomes observed with more real data. Figure 7(b) shows that a very short training time (10 epochs, 1hour on a single NVIDIA RTX 3090 GPU) leads to a rapid improvement in the accuracy of the synthetic classifier. When training ViT for 30 epochs, it achieves its highest accuracy, after which it starts to overfit. In contrast, CLIP pretrained from vast data shows no sign of overfitting.

## 4.2 REALTUNE SIGNIFICANTLY IMPROVES SYNTHETIC CLASSIFIERS IN CHALLENGING SCENARIOS

Next, we investigate the impact of RealTune on the challenging scenarios we identified in Section 2: fine-grained classification and multiple rare scenarios.

**Fine-grained classification.** Figure 2 shows that after RealTune, in the case of equal accuracy, the fine-grained classification accuracy of synthetic classifiers exceeds that of real classifiers (CLIP: 90.1% *v.s.* 88.75%, ViT: 88.21% *v.s.* 86.58%). In the case of equal data size, the gap between synthetic classifiers and real classifiers is further reduced. Notably, the difference between CLIP-Syn-371M and CLIP-Real-371M is only 0.9%.

Table 2: Performance of finetuned CLIP models on different rare scenarios of ImageNet-X and ImageNet. For RealTune w/ rare data, we use each type of rare data for finetuning, reporting the performance on the corresponding rare scenarios and the average accuracy of these five models on ImageNet.

| Model | Finetuning Strategy | ImageNet-X Performance (%) | | | | | ImageNet Acc (%) |
|---|---|---|---|---|---|---|---|
| | | Brighter | Darker | Larger | Smaller | Person | |
| CLIP-Real-371M | None | 5.8 | -15.2 | -4.3 | -18.2 | -36.3 | 66.77 |
| CLIP-Syn-371M | None | -3.7 | -18.4 | -7.8 | -30.6 | -38.6 | 55.68 |
| | RealTune w/ random data | 4.4 | -2.4 | -7.2 | -21.2 | -37.8 | **71.71** |
| | RealTun w/ rare data | **9.1** | **4.2** | **-5** | **10.1** | **-0.28** | 63.76 |

**Rare scenarios.** In this part, we consider two type of finetuning data for RealTune. The first type is randomly sampled real data from ImageNet. The second approach is more tailored down to the rare scenarios that synthetic classifiers struggle with (Section 2). Specifically, we use the quantitative metrics proposed in Section 3 to filter real data for each scenario (*e.g.,* brighter images). We report CLIP results in Table 2 and ViT results can be found in Table 5 in Appendix.

From Table 2, we can see that RealTune with random data enhances the robustness of the synthetic CLIP across these five scenarios and even surpasses real CLIP in "brighter" and "darker" scenarios, while RealTune with rare data achieves optimal performance in the corresponding scenarios. Nevertheless, RealTune with rare data still attains lower overall accuracy on ImageNet compared to random data, indicating that an emphasize on rare data may lead to a loss of data diversity. Similar conclusions hold for ViT result (see Table 5 in Appendix).

## 4.3 MIXED PRETRAINING FURTHER ENHANCES REALTUNE

Seeing the great benefits of RealTune, we ask whether mixing real and synthetic data during **pretraining** can also lead to improved performance. To answer this question, we randomly sample 100 images per class (totally 10k, 7.7% of ImageNet-100) from ImageNet-100 as real dataset, while the synthetic dataset comprises 100k images generated by Stable Diffusion. Following this, we train ResNet-18 and ViT-Tiny using different combinations of data for pretraining and finetuning stages, and the results are shown in Table 3 below. Refer to Appendix A.2 for the study on the mixing ratios of real data and synthetic data.

Table 3: Test accuracy on ImageNet-100 with different pretraining and finetuning data. The real data used in the two stages of "Mix-Real" is the same.

| Pretraining | Real | | Syn | | Mix | |
|---|---|---|---|---|---|---|
| Finetuning | None | Syn | None | Real | None | Real |
| ResNet | 45.8 | 47 | 48.6 | 63.3 | 64.8 | **65.8** |
| ViT | 44.7 | 41.7 | 40.6 | 48.4 | 50.9 | **54** |

We can see that the ranking of final performance is: Mix-Real > Mix-None > Syn-Real > Real-Syn, where Mix-Real stands for pretraining with mixed data and finetuning with real data. Mix-None and Syn-Real outperform using only synthetic data pretraining (Syn-None), suggesting that real data is advantageous in both the pretraining and finetuning phases. Consequently, Mix-Real, which uses

real data in both stages, achieves the best performance. In other words, a proper mixture of real and synthetic data can combine the best of both worlds to attain the optimal performance.

## 4.4 REALTUNE IN TEXT TASKS

The study above reflects the significant capability of RealTune in vision tasks, it provide valuable insights that may be applicable across various domains. Here, we investigate the impact of RealTune on GPT-2 (Radford et al., 2019) trained on synthetic data generated by GPT-4. The real data and synthetic data for pretraining both consist of 0.11M texts, and the finetune data size is 20% of the pretraining data. The models are pretrained for 15k steps and finetuned for 1k steps. The results are shown in Table 4. We observe that RealTune lead to a decrease in loss by 0.7 for GPT2-Syn, narrowing the gap with GPT2-Real, while SynTune resulted in a loss increase of 0.22 for GPT2-Syn. This indicates that RealTune is effective for text tasks as well.

Table 4: GPT-2 loss of different finetune methods. GPT-2-Real represents a model pretrained on real data, and GPT-2-Syn represents a model pretrained on an equivalent amount of synthetic data.

| Model | Vanilla | SynTune | RealTune |
|---|---|---|---|
| GPT2-Real | 2.78 | 3.21 | 3.04 |
| GPT2-Syn | 4.29 | 4.51 | 3.59 |

> **Takeaways of Section 4**
>
> The limitations of synthetic classifiers can be efficiently and effectively remedied by finetuning on a small set of real data, which we call **RealTune**. It drastically improves its overall performance as well as its capability at fine-grained classification and rare scenarios.

## 5 RELATED WORK

**Training from Synthetic data.** Many works (Islam et al., 2021; Huang et al., 2018; Wang et al., 2024) have explored training representation learning on synthetic data from various generative models. Bowles et al. (2018) and Bissoto et al. (2021) utilized images generated by GANs for medical diagnosis. Azizi et al. (2023) showed that data generated by diffusion models improved supervised learning by approximately 1% accuracy on ImageNet. Recently, text-to-image models have garnered widespread attention for visual representation learning. StableRep (Tian et al., 2023) treats various samples from the same real prompt as positives for contrastive learning, while SynCLR (Tian et al., 2024) replaces real prompts in StableRep with synthetic prompts from a large language model. Here, we focus on dissecting the gap between the real and synthetic classifiers and attributing the differences to the synthetic data. See Appendix C for more related works.

## 6 LIMITATIONS AND OUTLOOK

While our study provides valuable insights into the performance of synthetic classifiers in vision tasks, it is not without limitations. Firstly, our investigation is mainly confined to visual domains. Additionally, the range of challenging scenarios evaluated is limited; expanding this scope to include a more diverse set of conditions could offer a more comprehensive understanding of synthetic classifiers' capabilities. Furthermore, due to computational constraints, we were unable to perform retraining for each influencing factor individually, which might have yielded more detailed insights into the specific impacts of each element.

Looking ahead, future work could address these limitations by exploring widely the application of RealTune in other domains and incorporating a wider variety of challenging scenarios would enhance the robustness of our evaluations and provide a deeper understanding of synthetic classifiers' strengths and weaknesses. With increased computational resources, more granular studies could be conducted to isolate and examine the effects of different factors influencing classifier performance. Ultimately, these advancements will contribute to marrying synthetic and real data to foster the development of more resilient and versatile classifiers across multiple domains.

## REPRODUCIBILITY STATEMENT

The ViT and CLIP used for evaluation in our study are sourced from the checkpoints provided by SynRep (Fan et al., 2024). The Stable Diffusion (Rombach et al., 2022), Sing Diffusion (Zhang et al., 2024), and DeepFloyd IF (Shonenkov et al., 2023) used for data generation are obtained from official repositories, with details on data generation (prompts, CFG) described in detail in setup of Section 3. The YOLO V5 (Ultralytics, 2021) and YOLO World (Cheng et al., 2024) used for object detection, ConvNext-B (Liu et al., 2022) for studying Fine-grained class confusion, and BLIP-2 for prompt generation in Section 3 are all sourced from official repositories. The evaluation methods for each rare scenario are detailed in Section 3.2 and Appendix A.1. Experimental details for Section 4 are shown in setup of Section 4 and Appendix A.2. The datasets used in this study, including ImageNet (Deng et al., 2009), ImageNet-100 (Tian et al., 2020), ImageNet-V2 (Recht et al., 2019) and ImageNet-X (Idrissi et al., 2023), are all publicly available datasets provided by the official sources.

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

# A  EXPERIMENTAL DETAILS

## A.1  PROMPT DESIGN FOR RARE SCENARIOS

When investigating the impact of prompts on synthetic data, we adding phrases describing the specific rare scenarios in the default IN-Caption format prompts.

For object scale, as the default synthetic data images lack excessively large or small objects, we inserted "large" into one-third of the prompts (*e.g.,* "large tench, a man holding a fish"), "small" into another one-third (*e.g.,* "small wombat, an animal is standing on a log"), and left the remaining one-third unchanged.

For brightness, as the default synthetic data lack excessively bright or dark images, we inserted "in a bright environment" into one-third of the prompts (*e.g.,* "tench, a man holding a fish in a bright environment"), "in a dark environment" into another one-third, and left the remaining one-third unchanged.

For person blocking, we inserted "occluded by human body" into all prompts, *e.g.,* "tench occluded by human body, a man holding a fish in a bright environment".

## A.2  MODEL TRAINING SETTING

For the experiment in Figure 6(a), we finetune CLIP using 40k real images sampled from the ImageNet training set for 50 epochs, with a batch size of 128, 1000 warm-up steps, AdamW optimizer,and a learning rate of 5e-6 for RealTune. In SynTune, we utilize 40k synthetic images generated from Stable Diffusion and keep the other parameters consistent with RealTune.

For the experiment in Figure 6(b), we finetune ViT using 40k real images sampled from the ImageNet training set for 30 epochs, with a batch size of 256, SGD optimizer, and a learning rate of 3e-2 for RealTune. In SynTune, we utilize 40k synthetic images generated from Stable Diffusion and keep the other parameters consistent with RealTune.

For the experiment in Figure 7(a), we vary the data size for RealTune to be 13k, 40k, 52k, 65k, and 130k (corresponding to 1%, 3%, 5%, and 10% of ImageNet). Due to the changes in training data size, to ensure fair comparison, each scenario is trained for 6k steps. Other parameters remained consistent with those in Figure 6.

For the experiment in Figure 7(b), we adjust the training durations for RealTune to be 10, 20, 30, 50, and 100 epochs, while keeping other parameters consistent with those in Figure 6.

For the experiment in Table 3, we randomly sample 10k images from ImageNet100 as the real dataset, while the synthetic dataset consiste of 100k images generated by Stable Diffusion. We pretrain ResNet-18 for 100k steps with a batch size of 128, a learning rate of 0.1, and SGD optimizer. We finetune ResNet-18 for 2k steps with a batch size of 128, a learning rate of 1e-3, and SGD optimizer. We pretrain ViT-Tiny for 30k steps with a batch size of 512, a learning rate of 5e-4, AdamW optimizer, and 0.05 weight decay. We finetune ViT-Tiny for 3k steps with a batch size of 256, a learning rate of 1e-5, and SGD optimizer.

# B  ADDITIONAL RESULS

**ViT results at rare scenarios.** Figure 8 shows the evaluation results for ViT on ImageNet-X. It is observed that ViT-Syn-1M is less robust for larger, smaller, person blocking, multiple objects, brighter, style *etc.* at equal data size, while ViT-Syn-2M is less robust for larger, darker, multiple objects, and brighter *etc.* at equal accuracy. This conclusion is similar to CLIP in Section 2.1, further illustrating that synthetic classifiers face challenges with rare scenarios related to object scale, brightness, and complex scenarios like person blocking.

**Results of finetuning on ImageNet-V2.** In Section 4, we finetune models on randomly sampled data from ImageNet training set and evaluate on the ImageNet validation set. To avoid the benefits of in-domain data, we next finetune models on ImageNet-V2 and evaluate on ImageNet validation set. Since the amount of data in ImageNet-V2 is only 20k, we use 20k for finetuning in both RealTune and SynTune in Figure 9, unlike the 40k used in Section 4. As we can see in Figure 9, similar to the

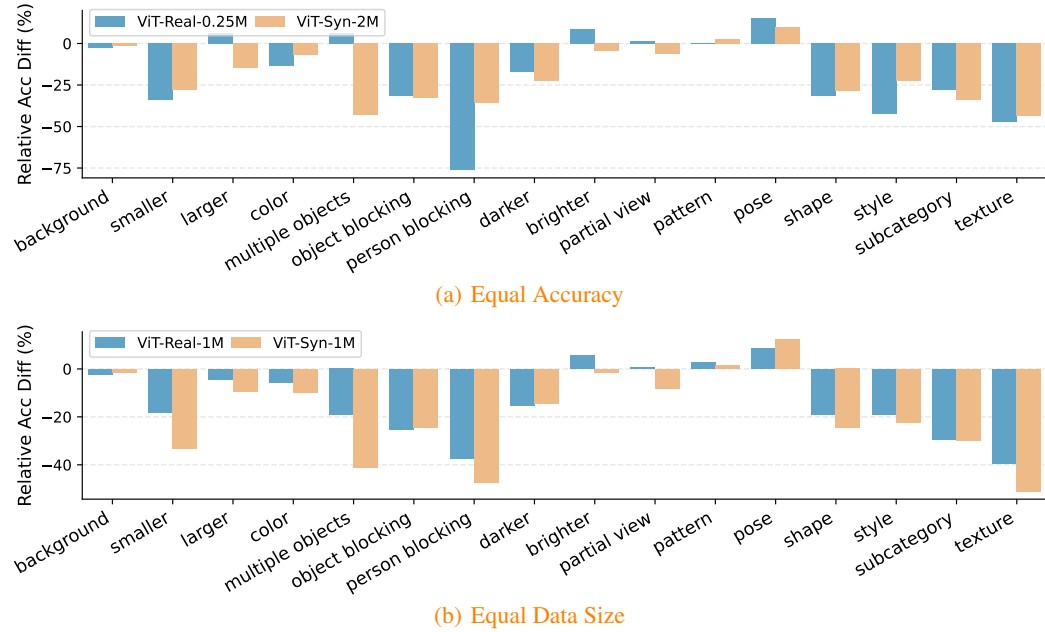

(a) Equal Accuracy

(b) Equal Data Size

Figure 8: Rare Scenario robustness of ViT on ImageNet-X.

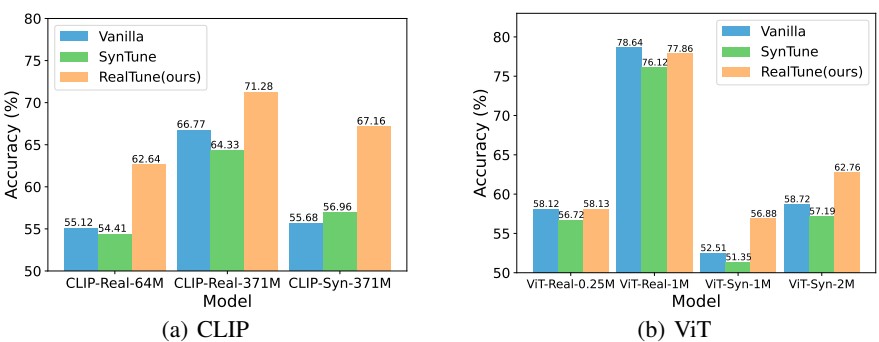

(a) CLIP

(b) ViT

Figure 9: ImageNet classification accuracy of different finetuning methods on ImageNet-V2. We utilize 20k images for finetuning in both RealTune and SynTune due to the total of 20k images in ImageNet-V2, unlike the 40k images used in Section 4.

results of finetuning on ImageNet data, RealTune significantly enhances the accuracy of synthetic classifiers. With RealTune, synthetic classifiers surpass real classifiers when pretraining accuracy is comparable. This further underscores the effectiveness and efficiency of RealTune in enhancing the performance of synthetic classifiers.

**ViT results of rare scenario robustness after Realtune.** Table 5 shows the robustness of synthetic ViT after RealTune with randomly sampled data and tailored rare data. We can see that as similar to CLIP results in Table 2, RealTune with random data enhances the robustness of the synthetic ViT across most scenarios and even surpasses real ViT in "larger" scenario, while RealTune with rare data achieves optimal performance in "brighter", "larger" and "smaller" scenarios. Nevertheless, RealTune with rare data still attains lower overall accuracy on ImageNet compared to random data, indicating that an emphasize on rare data may lead to a loss of data diversity.

**Comparison of different mixing methods.** Section 4.3 demonstrates the effectiveness of training with a mixture of real data and synthetic data. Here, we compare our mixing method with the mixing method proposed in He et al. (2023). For clarity, we refer to our method as "MixData" and the method from He et al. (2023) as "MixLoss." In "MixLoss," the losses of real data and synthetic data are summed at each iteration for backpropagation while our "MixData" combines real and synthetic data for training using the standard forward and backward methods. As shown in Table 6, accuracy

Table 5: Performance of finetuned ViT models on different rare scenarios of ImageNet-X and ImageNet. For RealTune w/ rare data, we use each type of rare data for finetuning, reporting the performance on the corresponding rare scenarios and the average accuracy of these five models on ImageNet.

| Model | Finetuning Strategy | ImageNet-X Performace (%) | | | | | ImageNet Acc (%) |
|---|---|---|---|---|---|---|---|
| | | Brighter | Darker | Larger | Smaller | Person | |
| ViT-Real-1M | None | 58 | -15.2 | -4.3 | -18.2 | **-37.5** | **78.64** |
| ViT-Syn-1M | None | -1.6 | **-14.5** | -9.5 | -33.2 | -47.5 | 52.51 |
| | RealTune w/ random data | 5.7 | -19.5 | -2.4 | -31 | -48.3 | 59.96 |
| | RealTun w/ rare data | **11.6** | -15.5 | **-0.2** | **-13.3** | -41 | 57.38 |

of MixData pretraining higher than that of MixLoss. After RealTune, the model performance of MixData further improves, while the model performance of MixLoss decreases. Therefore, our MixData outperforms MixLoss.

Table 6: Comparison of different mixing methods.

| Pretraining | MixData | | MixLoss | |
|---|---|---|---|---|
| Finietuning | None | Real | None | Real |
| ResNet | 64.8 | 65.8 | 60.34 | 59.9 |
| ViT | 50.9 | 54 | 50.46 | 49.92 |

**The impact of the ratio of real data and synthetic data.** In Section 4.3, we investigated the impact of using different datasets during the pretraining and finetuning stages on model performance, utilizing 10k real data and 100k synthetic data. Here, we conducted an ablation study on the ratio of real data to synthetic data. We adjust the quantity of real data to 10k, 15k, and 20k while maintaining the synthetic data constant at 100k for experiments on ResNet18 following the experimental settings setting in Section 4.3. The experimental results are presented in Table 7. First, the results shows that the rankings under different ratios of real to synthetic data are as follows: Mix-Real > Mix-None > Syn-Real > Real-Syn, which is consistent with our conclusion in Section 4.3. Second, the findings indicate that regardless of the quantity of real data, it is beneficial for mix training and a higher amounts of real data leading to better results. The results shows the generality of our approach.

Table 7: ResNet18 performance under different ratios of real and synthetic data.

| Real data num | Mix-Real | Mix-None | Syn-Real | Real-Syn |
|---|---|---|---|---|
| 10k | 65.8 | 64.48 | 63.3 | 47 |
| 15k | 69.86 | 65.94 | 64.46 | 51.46 |
| 20k | 70.84 | 69.1 | 64.28 | 57.3 |

**Ablation study of data augmentation.** Data augmentation techniques (*e.g.,* cropping, brightness adjustment, blocking) are highly relevant to the failure modes of synthetic classifiers discussed in Section 2. Therefore, we conducted a more detailed ablation study on augmentation to investigate the impact of augmentation techniques. We compare models without augmentation (vanilla), brightness adjustment augmentation, blocking augmentation, crop augmentation, and models with a combination of brightness adjustment, blocking, and cropping. The results are as shown in Table 8 and 9. The performance gap between models with and without augmentation is minimal, and the difference between SynTune and RealTune after using augmentation has not decreased. This indicates that augmentation cannot easily addressed synthetic data failure modes. What matters is RealTune, underscoring the importance of real data.

**DINO results of different finetuning methods.** To explore whether RealTune is suitable for self-supervised learning, we conduct experiments on DINO using ImageNet-100. The random subset used for RealTune consists of randomly selecting 100 images per class in ImageNet100 ( 7.7% of ImageNet100). The results are shown in Table 10. The results indicate that without RealTune,

Table 8: ViT-Syn-1M ImageNet accuracy under different augmentation methods

|         | Vanilla | Bright | Block | Crop | Bright+Block+Crop |
|---------|---------|--------|-------|------|-------------------|
| SynTune | 50.81 | 51.03 | 50.59 | 51.47 | 51.81 |
| RealTune | 58.86 | 59.03 | 59.15 | 59.96 | 59.95 |

Table 9: CLIP-Syn-371M ImageNet accuracy under different augmentation methods

|         | Vanilla | Bright | Block | Crop | Bright+Block+Crop |
|---------|---------|--------|-------|------|-------------------|
| SynTune | 58 | 58.60 | 57.26 | 57.41 | 55.48 |
| RealTune | 70.51 | 71.21 | 71.21 | 71.17 | 69.98 |

DINO-Syn exhibits an accuracy 32.72% lower than DINO-Real, which is a significant gap. However, after RealTune was applied to DINO-Syn, the performance gap between DINO-Syn and DINO-Real narrows to 7.62 % (DINO-Real Vanilla *v.s.* DINO-Syn RealTune acc), further highlighting the effectiveness of RealTune in self-supervised learning.

Table 10: ImageNet100 classification accuracy of different finetune methods on DINO. DINO-Real represents pretraining on Real ImageNet100, while DINO-Syn represents pretraining on synthetic ImageNet100.

|           | Vanilla | SynTune | RealTune |
|-----------|---------|---------|----------|
| DINO-Real | 68.2 | 39.28 | 63.54 |
| DINO-Syn | 35.48 | 32.96 | 60.58 |

**CLIP results of finetuning on Caltech101 and EuroSAT.** Section 4.1 evaluated different finetuning methods on ImageNet. Here, we conduct experiments using CLIP on Caltech101 and EuroSAT (a remote sensing image scene classification dataset) to examine the effectiveness of RealTune. The results are shown in Table 11 and 12. Consistent with the observations in Section 4.1, RealTune significantly improves the accuracy of synthetic classifiers on the Caltech101 and EuroSAT. Without RealTune, CLIP-Real-64M outperforms CLIP-Syn-371M on the Caltech101 and EuroSAT noticeably , but after RealTune, the performance of the two becomes comparable. Additionally, the performance gap between CLIP-Syn-371M and CLIP-Real-371M is further reduced. This further demonstrates the effectiveness of RealTune.

**RealTune with unbalanced datasets.** To investigate whether RealTune adapts well even if the real data is unbalanced, We sample 21 images per class for ImageNet classes 1-50, 23 images per class for classes 51-100, 25 images per class for classes 101-150, and so on, until 59 images per class for classes 951-1000, creating an unbalanced dataset. The total number of images is consistent with the balanced dataset in Section 4, at 40k images. Using this unbalanced dataset for RealTune with ViT and CLIP, the experimental results are shown in Table 13. The results show that unbalanced RealTune only decreases performance by 0.52% for ViT and 0.27% for CLIP compared to balanced RealTune. This indicates that RealTune works well even in challenging unbalanced scenarios.

**Examples of synthetic data.** For a concrete understanding, we provide examples of the synthetic data with different generative models in Figure 10. Overall, it can be seen that there is still a gap in quality between synthetic data and real data.

## C   ADDITIONAL RELATED WORKS

**Evaluation in challenging scenarios.** Evaluating classifiers in various challenging scenarios provides a more comprehensive understanding of their robustness and generalization capabilities beyond standard in-domain evaluation. Common challenging scenarios of ImageNet include ImageNet-C (Hendrycks and Dietterich, 2018), ImageNet-R (Hendrycks et al., 2021), ImageNet-Sketch (Wang et al., 2019), ObjectNet (Barbu et al., 2019), *etc.* Sarıyıldız et al. (2023) and Fan et al.

Table 11: Caltech101 classification accuracy of different finetune methods.

| Model | Baseline | SynTune | RealTune |
|---|---|---|---|
| CLIP-Real-64M | 87.85 | 86.67 | 88.98 |
| CLIP-Real-371M | 90.22 | 89.77 | 92.20 |
| CLIP-Syn-371M | 83.89 | 82.09 | 88.87 |

Table 12: EuroSAT classification accuracy of different finetune methods.

| Model | Baseline | SynTune | RealTune |
|---|---|---|---|
| CLIP-Real-64M | 46.11 | 45.26 | 93.44 |
| CLIP-Real-371M | 43.73 | 45.22 | 95.25 |
| CLIP-Syn-371M | 27.59 | 29.44 | 93.74 |

(2024) find that synthetic classifiers outperform real classifiers on ImageNet-Sketch and ImageNet-R. However, Singh et al. (2024) in their experiments on ImageNet-C and ImageNet-3DCC (Kar et al., 2022) show that synthetic classifiers are significantly less robust to common corruptions in images. In this work, we find that these benchmarks can be particularly helpful for understanding the limitations of synthetic classifiers, and we have designed a suite of quantitative measures for evaluating these factors in the training data, providing an objective and data-centric way for evaluating models under these challenging scenarios.

**Mixing Real and Synthetic Data.** Synthetic data has been widely used across various domains in data-scarce scenarios (Sankaranarayanan et al., 2018; Seib et al., 2020; Sun et al., 2021). Fan et al. (2024) demonstrate that the zero-shot classification capability of CLIP trained on a mix of data surpasses that of CLIP trained solely on real or synthetic data. Frid-Adar et al. (2018) find that using generated medical images for synthetic data augmentation enhances the CNN's performance in medical image classification. However, He et al. (2023) and Wang et al. (2024) find that the potential of synthetic data remains untapped or even harms model performance due to distribution shift. To overcome the limitations of synthetic data, He et al. (2023) employed real data to supervise the sampling process of the generative model. Wang et al. (2024) proposed an adaptive mixing strategy of real and synthetic data for contrastive self-supervised learning. In this work, we observe that simply finetuning with a small amount of real data can be a surprisingly efficient and effective remedy to improve model performance and enhance its robustness, thereby avoiding complex design and training processes.

Table 13: Comparison of the results between unbalanced RealTune and balanced RealTune.

|  | ViT-Syn-1M | CLIP-Syn-371M |
| --- | --- | --- |
| Vanilla | 52.51 | 55.68 |
| Balance RealTune | 59.96 | 71.17 |
| Unbalance RealTune | 59.44 | 70.9 |

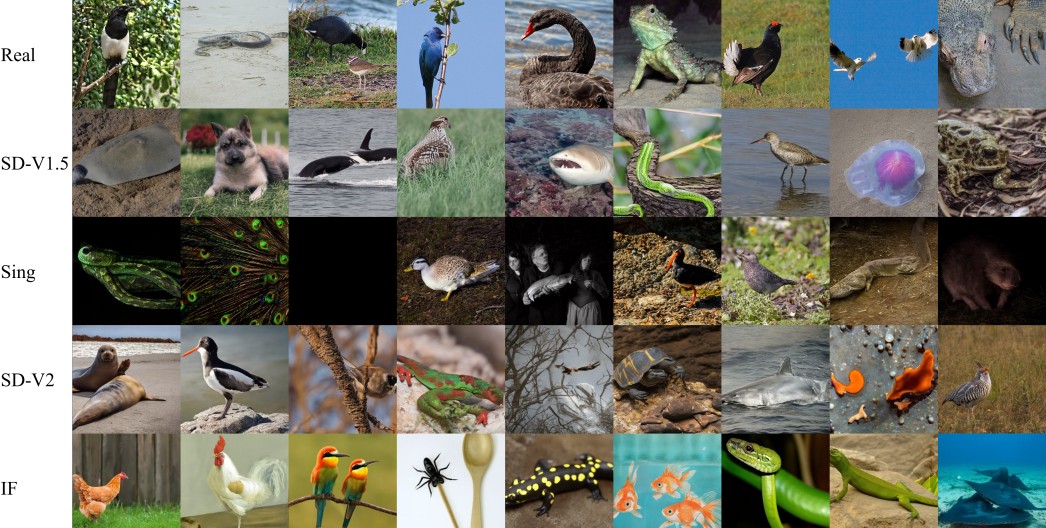

Figure 10: Real and synthetic data examples.

