# OpenReview forum: "Are Synthetic Classifiers Really as Good as Real Classifiers?"
_ICLR.cc/2025/Conference — Submitted to ICLR 2025_

### Official Review · Reviewer_rppm · 2024-10-15

**Soundness:** 2
**Presentation:** 3
**Contribution:** 2
**Rating:** 6
**Confidence:** 3

**Summary:**

The paper investigates whether classifiers trained on synthetic data can match the performance of those trained on real data, especially in visual tasks. Through comparative analysis, the authors find that although synthetic classifiers achieve similar overall accuracy to real-data-trained counterparts, they underperform in challenging scenarios such as fine-grained classification, extreme object scales, and brightness variations. The authors attribute these limitations to the inability of current generative models to fully capture the complexity and diversity of real-world data. To address these issues, they propose **RealTune**, a method that fine-tunes synthetic classifiers with a small amount of real data, significantly improving performance in these complex scenarios. The results demonstrate that combining synthetic and real data is essential to create more robust and efficient classifiers.

**Strengths:**

1. The paper provides a detailed comparison between classifiers trained on synthetic versus real data, especially in challenging scenarios like fine-grained classification and rare situations (e.g., extreme object scales, brightness variations, object occlusion). The authors use a range of quantitative metrics, such as employing YOLO models for object scale assessment and CIELAB color space for brightness evaluation, making their results systematic and reliable. This thorough analysis adds significant value to understanding the limitations of synthetic classifiers and potential improvements for future model development.
2. Generating synthetic data is relatively easy compared to collecting real-world data, but the quality and diversity often fall short. RealTune effectively addresses these shortcomings using minimal real data (around 3% of ImageNet), significantly enhancing the classifier’s performance. The proposed method also reduces computational and energy costs, making it particularly useful in resource-constrained environments. From a review perspective, this solution is practical, cost-effective, and demonstrates a clear advancement in addressing synthetic data challenges.
3. Beyond RealTune, the authors also explore the effectiveness of mixed data pretraining, showcasing the potential of combining real and synthetic data. This strategy not only improves classifier accuracy but also outperforms using either real or synthetic data alone. The experiments on mixed pretraining provide new insights into leveraging limited data resources effectively, which is crucial in scenarios where collecting large-scale real data is impractical. These contributions not only provide an empirical foundation for future research but also serve as valuable guidance for practical training strategies. The discussion of data-mixing approaches adds depth to the paper and offers an innovative direction for making optimal use of synthetic data.

**Weaknesses:**

1. The paper exclusively focuses on vision tasks without exploring other modalities like text or audio, which limits the generalizability of the findings. Given the growing relevance of synthetic data in various domains beyond vision (such as natural language and speech), this is a significant limitation. The authors briefly mention extending the study to other modalities in future work (Section 6), but as it stands, the narrow focus weakens the overall contribution. Extending the experiments to non-visual modalities or providing some preliminary analysis could have increased the paper’s broader applicability.
2. In the evaluation of rare scenarios, such as object occlusion, extreme object scales, and brightness variations, the authors use ImageNet-X (Figure 3). However, the analysis lacks an adequate diversity of datasets that could highlight other real-world challenges (e.g., dynamic backgrounds, motion blur, or domain shifts). ImageNet-X, though useful, does not fully encompass the variety of rare scenarios that might be seen in real-world applications. For a thorough evaluation, incorporating other benchmarks like ImageNet-C or ObjectNet could have provided a more comprehensive assessment of classifier robustness. This would also have helped to better validate the claims about synthetic data limitations.
3. Some figures in the paper are difficult to interpret due to suboptimal presentation choices. For example, Figure 4, which aims to demonstrate class consistency and frequency for fine-grained class confusion, uses bar plots that make it challenging to interpret the differences across models at a glance. The class consistency rates presented are not clearly distinguished, leading to potential confusion. Additionally, the visualization of synthetic data distribution shows class imbalance, but the representation could have been more effective with a clearer breakdown across different class labels to provide insight into how imbalance specifically affects model performance. The authors could benefit from using more readable visualizations like heatmaps or swarm plots that better convey the underlying relationships.
4. While RealTune demonstrates promising improvements, the experimental evaluation lacks adequate baseline comparisons against other established methods for enhancing synthetic classifiers. For example, the paper introduces SynTune as a counterpart to RealTune, but it would have been more informative to compare RealTune with other fine-tuning or data augmentation techniques that are popular in the field. Including a detailed comparison with transfer learning or data distillation methods could have added value to demonstrate RealTune's efficiency more convincingly. Moreover, while Figure 6 shows accuracy improvements, adding more baselines could help to better gauge the significance of the presented results.
5. The mixed training strategy, explored in Section 4.3 and illustrated in Table 3, shows that combining real and synthetic data during pretraining can enhance the performance of classifiers. However, the analysis lacks depth in explaining why certain combinations outperform others and how different ratios of real to synthetic data impact the results. The choice of using only 7.7% of real data (for ImageNet-100) in the mixed dataset is arbitrary and not well justified, limiting the ability to generalize findings to other datasets or settings. Additionally, there are no detailed ablation studies exploring various ratios between real and synthetic data, which would have provided a better understanding of the trade-offs and the optimal way to mix these data types. This omission reduces the experimental rigor and leaves questions regarding the optimal strategy for practical scenarios.

**Questions:**

1.*How well does RealTune generalize across different visual datasets or domains?*
RealTune was tested on ImageNet and its smaller subset, ImageNet-100, which are certainly widely used benchmarks. But I wonder—would the same level of improvement hold if we used a dataset with a lot more variation, like COCO, or even a domain-specific dataset like medical imagery? These types of datasets have different complexities, like intricate scene layouts or highly specialized objects, which might expose weaknesses that did not appear with ImageNet. It would be helpful to know if RealTune is a broadly applicable method for any kind of visual domain or if its effectiveness is more specific to the characteristics of ImageNet.

2.*How does the quality of the fine-tuning data affect RealTune's success?*
The authors showed that RealTune works well with a small amount of real data, but it made me wonder about the specifics of that fine-tuning data. What if the real data used for fine-tuning is biased or lacks diversity? Would RealTune still perform as well? Real-world data collection often has biases, like unbalanced classes or limited examples of certain challenging conditions, and it would be useful to understand how that affects the final model. Does RealTune require a carefully balanced and curated fine-tuning set, or can it adapt well even if the real data is subpar? It would be great to see more on whether the composition of this fine-tuning data matters.

3.*How does RealTune's computational efficiency compare with other methods in terms of energy use and scalability?*
RealTune was described as efficient, running on a single GPU in a relatively short time. But I would like to know more about how it stacks up against other methods in terms of energy usage, especially considering the push for more environmentally friendly AI solutions. How does RealTune compare, for example, to larger pretraining strategies or other fine-tuning techniques when it comes to energy consumption or the practicality of scaling to larger models? Understanding these trade-offs would be really important for someone trying to decide between RealTune and other enhancement methods, especially in scenarios with limited computational resources. It would also help gauge if the method's efficiency benefits hold up when scaling to bigger models or if there are diminishing returns.

---

> ### Author Response · Authors · 2024-11-22
> **Response to Reviewer rppm**
>
> Thanks for your detailed reading and pointing out problems of our work. We have revised the paper carefully following your suggestions. Below, we address your main concerns on the paper content.
>
> ---
>
> Q1: Extending the experiments to non-visual modalities or providing some preliminary analysis could have increased the paper’s broader applicability.
>
> A1: Thank you for pointing that out. Indeed, this paper focuses mainly on vision tasks, but understanding the effectiveness of a small amount of real data for synthetic data representation learning can still **provide valuable insights** that may be applicable across various domains. Here, we investigated the impact of RealTune on **GPT2** trained on synthetic data generated by GPT4. The real data and synthetic data for pretraining both consist of 0.11M texts, and the finetune data size is 20% of the pretraining data. The models are pretrained for 15k steps and finetuned for 1k steps.
>
> The results are shown in the table below. We observe that RealTune lead to **a decrease in loss by 0.7 for GPT2-Syn**, **narrowing the gap with GPT2-Real,** while SynTune resulted in a loss increase of 0.22 for GPT2-Syn. This indicates that RealTune is effective for text tasks as well, demonstrating its generalization.
>
> Experimental results are added in **Section 4.4**.
>
> *GPT2 loss of different finetune methods. GPT2-Real represents a model pretrained on real data, and GPT2-Syn represents a model pretrained on an equivalent amount of synthetic data.*
>
> |  | Vanilla | SynTune | RealTune |
> | --- | --- | --- | --- |
> | GPT2-Real | 2.78 | 3.21 | 3.04 |
> | GPT2-Syn | 4.29 | 4.51 | 3.59 |
> ---
> Q2: The analysis lacks an adequate diversity of datasets that could highlight other real-world challenges. ImageNet-X, though useful, does not fully encompass the variety of rare scenarios that might be seen in real-world applications. For a thorough evaluation, incorporating other benchmarks like ImageNet-C or ObjectNet could have provided a more comprehensive assessment of classifier robustness.
>
> A2: Considering evaluation on ImageNet-C is reasonable and valuable. In fact, **paper [1] has compared the performance of the synthetic and real classifiers (ViT and CLIP) on ImageNet-C** **and ImageNet-3DCC** (which includes 12 common corruptions that take depth into account, e.g., z-axis blur, far and near focus errors, etc.). Their results show that **the performance of synthetic classifiers on the two datasets is significantly lower than that of real classifiers, highlighting the limitations of synthetic data.** This has been mentioned in the related work of our paper.
>
> Through evaluation on ImageNet-X, we identified challenges for synthetic classifiers that are more relevant to real-world scenarios and model semantics. We added the discussion regarding ImageNet-C from paper[1] into **Section 2.1**.
>
> **Reference:**
>
> [1] Krishnakant Singh, Thanush Navaratnam, Jannik Holmer, Simone Schaub-Meyer, and Stefan Roth. Is synthetic data all we need? benchmarking the robustness of models trained with synthetic images. In CVPR, 2024.
>
> ---
>
> Q3: Some figures in the paper are difficult to interpret due to suboptimal presentation choices.
>
> A3: Thank you for your suggestions. We made slight adjustments to Figure 4(a) and **added a more detailed caption.** Additionally, we **replaced Figure 4(b) from a predicted label histogram to a fine-grained confusion matrix of 8 snake species** for a more intuitive understanding of fine-grained class confusion in the revision.

---

> > ### Author Response · Authors · 2024-11-22
> > **Response to Reviewer rppm**
> >
> > ---
> >
> > Q4: While RealTune demonstrates promising improvements, the experimental evaluation lacks adequate baseline comparisons against other established methods for enhancing synthetic classifiers. For example, the paper introduces SynTune as a counterpart to RealTune, but it would have been more informative to compare RealTune with other fine-tuning or data augmentation techniques that are popular in the field.
> >
> > A4: Thank you for pointing this out. To conduct a more rigorous evaluation, we incorporated different augmentation techniques during fine-tuning, which are highly relevant to the failure modes of synthetic classifiers revealed in Section 2, for comparison. **We compare models without augmentation (vanilla), brightness adjustment augmentation, blocking augmentation, crop augmentation, and models with a combination of brightness adjustment, blocking, and cropping.** The results are as follows. The performance gap between models with and without augmentation is minimal, and the difference between SynTune and RealTune after using augmentation has not decreased. This indicates that augmentation cannot easily addressed synthetic data failure modes. **What matters is RealTune**, underscoring the importance of real data.
> >
> > The results are added to **Appendix B**.
> >
> > *ViT-Syn-1M ImageNet accuracy under different augmentation methods*
> >
> > | Syn ViT-1M | Vanilla | Bright | Block | Crop | Bright+Block+Crop |
> > | --- | --- | --- | --- | --- | --- |
> > | SynTune | 50.81 | 51.03 | 50.59 | 51.47 | 51.81 |
> > | RealTune | 58.86 | 59.03 | 59.15 | 59.96 | 59.95 |
> >
> > *CLIP-Syn-371M ImageNet accuracy under different augmentation methods*
> >
> > | Syn CLIP-371M | Vanilla | Bright | Block | Crop | Bright+Block+Crop |
> > | --- | --- | --- | --- | --- | --- |
> > | SynTune | 58 | 58.06 | 57.26 | 57.41 | 55.48 |
> > | RealTune | 70.51 | 71.21 | 71.21 | 71.17 | 69.98 |
> >
> > ---
> >
> > Q5: The mixed training strategy, explored in Section 4.3 and illustrated in Table 3, shows that combining real and synthetic data during pretraining can enhance the performance of classifiers. However, (1) the analysis lacks depth in explaining why certain combinations outperform others and (2) how different ratios of real to synthetic data impact the results.
> >
> >  (3) There are no detailed ablation studies exploring various ratios between real and synthetic data, which would have provided a better understanding of the trade-offs and the optimal way to mix these data types.
> >
> > A5:
> >
> > > the analysis lacks depth in explaining why certain combinations outperform others
> > >
> >
> > We observe that Mix-None (pretraining with mixed data and no finetuning) and Syn-Real (pretraining with mixed data and finetuning with real data) outperform using only synthetic data pretraining （Syn-None）, suggesting that real data is advantageous in both the pretraining and finetuning phases. Consequently, Mix-Real, which uses real data in both stages, achieves the best performance. This analysis has been added into **Section 4.3.**
> >
> > > how different ratios of real to synthetic data impact the results
> > >
> >
> > Investigating the impact of different ratio of real and synthetic data on the results is highly worthwhile. Therefore, we **adjust the quantity of real data to 10k, 15k, and 20k** while maintaining the synthetic data constant at 100k for experiments on ResNet18 following the experimental settings setting in Section 4.3. The experimental results are presented in the table below. These experiments have been added into **Appendix B**.
> >
> > The results shows that the rankings under different ratios of real to synthetic data are as follows: **Mix-Real > Mix-None > Syn-Real > Real-Syn**, which **is consistent with our conclusion in Section 4.3**.
> >
> > > There are no detailed ablation studies exploring various ratios between real and synthetic data, which would have provided a better understanding of the trade-offs and the optimal way to mix these data types.
> > >
> >
> > The results indicate that **regardless of the quantity of real data, it is beneficial for mix training and a higher amounts of real data leading to better results.** The results shows the generality of our approach. Additionally, it is important to note that **section 4.3 of our paper is not aimed at finding an optimal mixing strategy but rather emphasizes the crucial role of real data in pretraining or fine-tuning** while many researchers are focusing on only utilizing synthetic data for model training.
> >
> > *Model performance under different ratios of real and synthetic data. Mix-Real represents pretraining with mixed data and finetuning with real data*
> >
> > | Real data num | Mix-Real | Mix-None | Syn-Real | Real-Syn |
> > | --- | --- | --- | --- | --- |
> > | 10k | 65.8 | 64.48 | 63.3 | 47 |
> > | 15k | 69.86 | 65.94 | 64.46 | 51.46 |
> > | 20k | 70.84 | 69.1 | 64.28 | 57.3 |
> >
> > ---

---

> > > ### Author Response · Authors · 2024-11-22
> > > **Response to Reviewer rppm**
> > >
> > > Q6: *How well does RealTune generalize across different visual datasets or domains?*
> > >
> > > A6: Thank you for pointing this out. We conduct experiments using CLIP on **Caltech101 and EuroSAT** (a remote sensing image scene classification dataset) . The results are shown in tables below. Consistent with the observations in Section 4.1 of our paper on ImageNet experiments, **RealTune significantly improves the accuracy of synthetic classifiers on the Caltech101 and EuroSAT.** Without RealTune, CLIP-Real-64M outperforms CLIP-Syn-371M on the Caltech101 and EuroSAT noticeably , but **after RealTune, the performance of the two becomes comparable.** Additionally, the performance gap between CLIP-Syn-371M and CLIP-Real-371M is further reduced. This further demonstrates the effectiveness of RealTune. These experiments have been added into **Appendix B.**
> > >
> > > *Caltech101 classification accuracy of different finetune methods*
> > >
> > > | Model | Baseline | SynTune | RealTune |
> > > | --- | --- | --- | --- |
> > > | CLIP-Real-64M | 87.85 | 86.67 | 88.98 |
> > > | CLIP-Real-371M | 90.22 | 89.77 | 92.20 |
> > > | CLIP-Syn-371M | 83.89 | 82.09 | 88.87 |
> > >
> > > *EuroSAT classification accuracy of different finetune methods*
> > >
> > > | EuroSAT | Baseline | SynTune | RealTune |
> > > | --- | --- | --- | --- |
> > > | CLIP-Real-64M | 46.11 | 45.26 | 93.44 |
> > > | CLIP-Real-371M | 43.73 | 45.22 | 95.25 |
> > > | CLIP-Syn-371M | 27.59 | 29.44 | 93.74 |
> > >
> > > ---
> > >
> > > Q7: *How does the quality of the fine-tuning data affect RealTune's success?* What if the real data used for fine-tuning is biased or lacks diversity? Would RealTune still perform as well?  Does RealTune require a carefully balanced and curated fine-tuning set, or can it adapt well even if the real data is subpar? It would be great to see more on whether the composition of this fine-tuning data matters.
> > >
> > > A7:  We explored the performance of RealTune in  unbalanced scenario.
> > >
> > > **Setup.** We sample 21 images per class for ImageNet classes 1-50, 23 images per class for classes 51-100, 25 images per class for classes 101-150, and so on, until 59 images per class for classes 951-1000, creating an unbalanced dataset. The total number of images is consistent with the balanced dataset in Section 4, at 40k images.
> > >
> > > The results show that unbalanced RealTune only decreases performance by 0.52% for ViT and 0.27% for CLIP compared to balanced RealTune. **This indicates that RealTune works well even in challenging unbalanced scenarios.** Moreover, since RealTune requires a very small amount of data (just 40 images per class for ImageNet), **it is easy to collect a high-quality small dataset for RealTune.** Therefore, utilizing a large but low-quality synthetic dataset for pretraining and then finetuning with a high-quality small real dataset is a very promising algorithm. The results are added in **Appendix B**.
> > >
> > > *Comparison of the results between unbalanced RealTune and balanced RealTune*
> > >
> > > |  | ViT-Syn-1M | CLIP-Syn-371M |
> > > | --- | --- | --- |
> > > | Vanilla (no finetune) | 52.51 | 55.68 |
> > > | Balance RealTune | 59.96 | 71.17 |
> > > | Unbalance RealTune | 59.44 | 70.9 |
> > >
> > > ---
> > >
> > > *Q8:* *How does RealTune's computational efficiency compare with other methods in terms of energy use and scalability?* How does RealTune compare, for example, to larger pretraining strategies or other fine-tuning techniques when it comes to energy consumption or the practicality of scaling to larger models?
> > >
> > > A8: Compared with model pretraining, RealTune offers significant advantages in terms of resource and time consumption. For example, ViT pretraining on ImageNet requires 3 days on 4 GPUs [1], whereas RealTune only needs 1 hour on a single GPU.
> > >
> > > [1] Touvron, Hugo, et al. "Training data-efficient image transformers & distillation through attention." *International conference on machine learning*. PMLR, 2021.
> > >
> > > ---
> > >
> > > Thank you again for your careful reading. We have carefully refined our paper following your suggestions, and addressed each of your concerns above. We respectfully suggest that you could re-evaluate our work based on the updated results. We are very happy to address your remaining concerns on our work.

---

> > > > ### Comment · Reviewer_rppm · 2024-11-22
> > > >
> > > > Thanks for your detail rebuttal, I think most of my concerns are solved, I will change the score.

---

> > > > > ### Author Response · Authors · 2024-11-22
> > > > >
> > > > > Thank you for increasing the score! We are glad to see that our responses resolved your concerns. Have a great day!

---

### Official Review · Reviewer_5XMN · 2024-11-01

**Soundness:** 2
**Presentation:** 3
**Contribution:** 2
**Rating:** 3
**Confidence:** 5

**Summary:**

This paper investigated the performance of real classifiers and synthetic classifiers. In particular, the authors claim that synthetic classifiers exhibit deficiencies in a range of challenging real-world scenarios, such as fine-grained classification, extreme object scales and extreme brightness despite achieving comparable overall accuracy to their real-data-trained counterparts. To this end, the authors proposed RealTune, a method that enhances synthetic classifiers by finetuning them with a small amount of real data. Experimental evaluations demonstrate that RealTune significantly improves the performance of synthetic classifiers using only a limited real dataset.

**Strengths:**

This paper is well presented and easy to follow.

**Weaknesses:**

In general, this paper conducted both analysis on synthetic classifiers and proposed an approach, however both parts require tremendous extra work to improve for a solid contribution.

* a) The claim that synthetic classifiers perform worse on fine-grained dataset and rare scenarios is not well justified. From Tab. 1 and Fig. 2, synthetic classifiers and real classifiers show similar trend on both IN results and fine-grained results. Synthetic 371M shows very similar accuracy as real 64M. This is the case for both regular IN images and fine-grained images. In Fig. 3, the so called relative accuracy difference is 0.5%, which does not show significance. Not to mention that there are positive values, indicating poor consistency.

* b) The fact that generative models fail to generate fine grained is simply not true. There are many works that focus on generating images in the context of fine-grained images. For instance, SphericGAN: Semi-supervised Hyper-spherical Generative Adversarial Networks for Fine-grained Image Synthesis; Semi-Supervised Single-Stage Controllable GANs for Conditional Fine-Grained Image Generation; The authors should study the synthetic classifiers with data generated with models designed specifically for Fine-Grained Images.

* c) There is no comparison conducted w.r.t prior arts that work with training models using mixture of synthetic data and real data. As surveyed by the authors themselves in the related work section, yet the authors compared with none of them.

* d) Training models with both synthetic data and real data is simply not novel at all.

**Questions:**

See weakness.

---

> ### Author Response · Authors · 2024-11-22
> **Response to Reviewer 5XMN**
>
> Thank you for your careful reading and pointing out our problems! We have carefully revised the paper following your suggestions. We address your concerns as follows.
>
> ---
>
> Q1: The claim that synthetic classifiers perform worse on fine-grained dataset and rare scenarios is not well justified.
>
> (1) From Tab. 1 and Fig. 2, synthetic classifiers and real classifiers show similar trend on both IN results and fine-grained results. Synthetic 371M shows very similar accuracy as real 64M.
>
> (2) In Fig. 3, the so called relative accuracy difference is 0.5%.
>
> A1：
>
> > From Tab. 1 and Fig. 2, synthetic classifiers and real classifiers show similar trend on both IN results and fine-grained results. Synthetic 371M shows very similar accuracy as real 64M.
> >
>
> (1) As you have mentioned, it is evident that synthetic classifiers and real classifiers exhibit a similar trend in both IN results and fine-grained results. **This similar trends highlights that the inferior performance of synthetic classifiers (IN results) can be mainly attributed to their tendency to confuse closely related subclasses (lower fine-grained accuracy).**
>
> For CLIP-Real-64M v.s. CLIP-Syn-371M, the checkpoints we used from paper[1]. And paper[1] did not report the models' variance or provide multiple checkpoints for us to calculate variance. And it was challenging for us to re-pretrain CLIP due to the computational demands of generating synthetic data and pretraining. However, it is generally believed that the variance at the ImageNet level is within 0.3. Therefore, we consider the differences of 83.5 vs. 83.1 (CLIP-Real-64M v.s. CLIP-Syn-371M) to be significant. Additionally, at an equal data size, the gap of fine-grained results between real and synthetic classifiers are more significant.
>
> **References:**
>
> [1] Lijie Fan, Kaifeng Chen, Dilip Krishnan, Dina Katabi, Phillip Isola, and Yonglong Tian. Scaling
> laws of synthetic images for model training... for now. In CVPR, 2024.
>
> > *In Fig. 3, the so called relative accuracy difference is 0.5%.*
> >
>
> (2) Regarding the statement "In Fig. 3, the so-called relative accuracy difference is 0.5%," we  find **we made an error in labeling the y-axis scale during plotting Fig. 3. It should have been 50% instead of 0.5%.** From the modified Fig. 3, we can see that synthetic classifiers perform significantly worse than real classifiers in certain rare scenarios. **Taking CLIP at equal data size as an example,** CLIP-Syn-371M performed **15% lower** on darker images, **20% lower** on smaller objects, and **40% lower** on person blocking compared to CLIP-Real-371M as shown in Figure 3. Thank you for pointing out this important mistake. We have corrected Fig. 3 in the revision.
>
> ---
>
> Q2: The fact that generative models fail to generate fine grained is simply not true. There are many works that focus on generating images in the context of fine-grained images. For instance, SphericGAN: Semi-supervised Hyper-spherical Generative Adversarial Networks for Fine-grained Image Synthesis; Semi-Supervised Single-Stage Controllable GANs for Conditional Fine-Grained Image Generation; The authors should study the synthetic classifiers with data generated with models designed specifically for Fine-Grained Images.
>
> A2: Regarding the potential for better models or specialized methods to address the issue of fine-grained classes is a very nice suggestion. In fact, in our paper, we also attempted various approaches to enhance the generation capabilities of fine-grained classes, such as adjusting classifier-free guidance, modifying prompts, and switching to better generative models. However, as shown in Figure 4a, increasing the CFG scale can improve class consistency but still falls short of closing the gap with real data. And increasing CFG will lead to synthetic datasets lack of diversity, resulting in a decrease in the synthetic classifier's performance. Therefore, **this method-specific approach does not entirely resolve the issue, as while it may improve this particular problem, it could result in a decline in other aspects.**
>
> Regarding the two papers you mentioned, as they do not provide code, it is challenging for us to reproduce these experiments within the limited time. If you have access to the code, we would be happy to compare our results with theirs. Anyway, we believe that, **just as we have attempted in Figure 4a, if they may excel in fine-grained aspect, they might fall short in other aspects (e.g., diversity).**
>
> ---

---

> > ### Comment · Reviewer_5XMN · 2024-11-22
> >
> > Thank you for your response, yet I am not convinced by the authors' response. In particular:
> >
> > * Given a possible variance of 0.3, an accuracy comparison of 83.5 vs. 83.1 does not justify the claim that synthetic classifiers perform worse on fine-grained dataset and rare scenarios.
> >
> > * Assume that improving quality of generated fine-grained images would indeed solve the problem, using images generated by approaches designed specifically for fine-grained would be a simple solution while staying in the domain of synthetic images rather than introducing real images. However, this was neglected by the authors.
> >
> > * Mixing real data with synthetic data shows limited novelty.
> >
> > I will thus retain my original rating.

---

> > > ### Author Response · Authors · 2024-11-23
> > > **Response to Reviewer 5XMN**
> > >
> > > Thanks for your feedback. We will address your remaining concern below.
> > >
> > > ---
> > >
> > > Q1: Given a possible variance of 0.3, an accuracy comparison of 83.5 vs. 83.1 does not justify the claim that synthetic classifiers perform worse on fine-grained dataset and rare scenarios.
> > >
> > > A1: We are sorry for causing this confusion. In fact, we were trying to say that ImageNet accuracy's variation is usually **at most 0.3** (**not its variance**). This is a common practice in ImageNet-scale experiments, and in real world, the variance/stdev is usually much smaller. For example, in ViT paper [1], they report 85.30 ± 0.02 for ViT and 87.54 ± 0.02 for ResNet (Table 2 [1]). Therefore, we believe that a difference by 0.4 can be considered as a clear difference on ImageNet. Though, we do acknowledge that it is not a very large gap, and the gap is clearer when comparing equal data sizes. We have added this to the footnote to avoid misunderstanding. Thank you for bringing it up to us, and please let us know if there is more to clarify. We appreciate your engagement and we are happy to address them in the remaining period.
> > >
> > > **References:**
> > >
> > > [1] Alexey Dosovitskiy, Lucas Beyer, Alexander Kolesnikov, Dirk Weissenborn, Xiaohua Zhai, Thomas
> > > Unterthiner, Mostafa Dehghani, Matthias Minderer, Georg Heigold, Sylvain Gelly, Jakob Uszkoreit, and Neil Houlsby. An image is worth 16x16 words: Transformers for image recognition at scale. In ICLR, 2021.
> > >
> > > ---
> > >
> > > Q2: Assume that improving quality of generated fine-grained images would indeed solve the problem, using images generated by approaches designed specifically for fine-grained would be a simple solution while staying in the domain of synthetic images rather than introducing real images. However, this was neglected by the authors.
> > >
> > > A2: Improving generative models to improve the quality of fine-grained images may not be a "simple solution". The results in Figure 4a show that the fine-grained problem cannot be resolved by **simply** increasing CFG, modifying prompts, or switching to better generative models. **It may require larger model size and computational resources to improve generative models, which increases costs.**  In contrast, **a more cost-effective and efficient approach is fine-tuning with a small amount of real data.** As shown in Figure 2, before RealTune, the fine-grained accuracy of ViT-Syn-2M was lower than ViT-Real-0.25M; however, after RealTune, ViT-Syn-2M outperformed ViT-Real-0.25M by 1.63%. The cost for this is only 40k images and training for an hour on a single  NVIDIA RTX 3090 GPU.
> > >
> > > ---
> > >
> > > Q4: Mixing real data with synthetic data shows limited novelty.
> > >
> > > A: In fact, mix training has been studied by many works. However, the purpose of our mix training experiment is not to design a new method **but to explore whether real data can also play an important role in the model synthetic data pretraining stage** just like real data in the synthetic classifier finetuning stage (RealTune). The results shows that real data is also important in model pretraining. So using real data (called Mix-Real in our paper) in both pretraining and finetuning stages can yield the best model performance.
> > >
> > > ---
> > >
> > > Thank you again and hope the clarification above could address your concerns. Please let us know if there is more to clarify.

---

> > > > ### Author Response · Authors · 2024-11-25
> > > >
> > > > Dear Reviewer 5XMN,
> > > >
> > > > Thank you for your thorough review and feedback once again.
> > > >
> > > > We have provided detailed answers to the concerns you still have. Would you please take a look and let us know whether you find it satisfactory?
> > > >
> > > > If you still have any questions, please feel free to ask.
> > > >
> > > > Thanks! Have a great day!
> > > >
> > > > Authors

---

> ### Author Response · Authors · 2024-11-22
> **Response to Reviewer 5XMN**
>
> Q3：There is no comparison conducted w.r.t prior arts that work with training models using mixture of synthetic data and real data. As surveyed by the authors themselves in the related work section, yet the authors compared with none of them.
>
> A3: Thanks you for reminding us. In the revision, we **added a comparison with the method  described in paper [1]** which mentioned in our related work section. For clarity, we refer to our method as "MixData" and the method from paper [1] as "MixLoss." In "MixLoss," the losses of real data and synthetic data are summed at each iteration for backpropagation while our "MixData" combines real and synthetic data for training using the standard forward and backward methods. **As shown in the revision (quoted below), accuracy of MixData pretraining higher than that of MixLoss. After RealTune, the model performance of MixData further improves, while the model performance of MixLoss decreases. Therefore, our MixData outperforms MixLoss.** The results are added in **Appendix B.**
>
> For other related work [2-7], we observe that they are specifically tailored towards particular directions or models, making them unsuitable for direct comparison with our approach. For instance, [2] is tailored for contrastive learning by reducing the augmentation strength for generating positive and negative samples, [3] for semantic segmentation, [4] for optical flow, [5] for the medical field, [6] for CLIP pretraining, and [7] primarily for traffic object detection. Nevertheless, investigations across diverse domains [2-7] also underscore the significance of exploring the mixing of real and synthetic data.
>
> *Comparison of different mixing methods. MixData is our method, and MixLoss is from paper [1]. MixData+RealTune (MixLoss+RealTune) refers to training the model with MixData (MixLoss) and then fine-tuning it with real data.*
>
> |  | MixData (ours) | MixData+RealTune (ours) | MixLoss | MixLoss+RealTune |
> | --- | --- | --- | --- | --- |
> | ResNet | 64.8 | 65.8 | 60.34 | 59.9 |
> | ViT | 50.9 | 54 | 50.46 | 49.92 |
>
> **References**
> [1] Ruifei He, Shuyang Sun, Xin Yu, Chuhui Xue, Wenqing Zhang, Philip Torr, Song Bai, and XI-
> AOJUAN QI. Is synthetic data from generative models ready for image recognition? In ICLR,
> 2023.
>
> [2] Yifei Wang, Jizhe Zhang, and Yisen Wang. Do generated data always help contrastive learning? In ICLR, 2024.
>
> [3] Swami Sankaranarayanan, Yogesh Balaji, Arpit Jain, Ser Nam Lim, and Rama Chellappa. Learning from synthetic data: Addressing domain shift for semantic segmentation. In CVPR, 2018.|
>
> [4] Deqing Sun, Daniel Vlasic, Charles Herrmann, Varun Jampani, Michael Krainin, Huiwen Chang, Ramin Zabih, William T Freeman, and Ce Liu. Autoflow: Learning a better training set for optical flow. In CVPR, 2021.
>
> [5] Maayan Frid-Adar, Idit Diamant, Eyal Klang, Michal Amitai, Jacob Goldberger, and Hayit
> Greenspan. Gan-based synthetic medical image augmentation for increased cnn performance
> in liver lesion classification. Neurocomputing, 321:321–331, 2018
>
> [6] Lijie Fan, Kaifeng Chen, Dilip Krishnan, Dina Katabi, Phillip Isola, and Yonglong Tian. Scaling
> laws of synthetic images for model training... for now. In CVPR, 2024.
>
> [7] Viktor Seib, Benjamin Lange, and Stefan Wirtz. Mixing real and synthetic data to enhance neural network training–a review of current approaches. arXiv preprint arXiv:2007.08781, 2020
>
> ---
>
> Q4：Training models with both synthetic data and real data is simply not novel at all.
>
> A4: Using both synthetic data and real data for model training has indeed been explored in many studies. However, the design for using both is not the primary focus of our paper. **What we aim to investigate is whether real data can bridge the significant gap between synthetic and real classifiers, and at what cost this gap can be bridged.** Our research in Section 4 reveals that by **using a very small amount of real data for a short finetuning period**, we can rapidly bridge this gap, a finding that has not been revealed in previous articles.
>
> ---
>
> Thank you again for your careful reading. We have carefully refined our paper following your suggestions, and addressed each of your concerns above. We respectfully suggest that you could re-evaluate our work based on the updated results. We are very happy to address your remaining concerns on our work.

---

### Official Review · Reviewer_URhH · 2024-11-03

**Soundness:** 3
**Presentation:** 4
**Contribution:** 2
**Rating:** 5
**Confidence:** 4

**Summary:**

This paper performs an analysis of pretraining on synthetic generated images. The authors break down some ways in which these models fall short and how this is reflected in the limits of image generators. For example, an off-the-shelf image generator does not produce images with much variety in image brightness so a classifier trained on these images tends to do worse on very dark or very bright images. Similarly, the image generator will make mistakes and mix up fine-grained semantic categories which leads to noisier supervision.

The authors discuss ways to potentially address these issues when generating images (attempting to adjust settings and prompts) but show there is little to be done on that front. What proves more effective is finetuning on a small set of randomly sampled real data (40k images). It is also helpful to pretrain on a mix of real and synthetic data.

**Strengths:**

This exposition of this paper is quite good, the authors point to very specific limitations and take us through their process to address them. There are quite a number of interesting and detailed analyses sprinkled through the paper, some in particular that stand out to me:
- I like the thought that went into Section 3, and the various ways to pick at how good a generative model is at producing reasonable training data (e.g. the class consistency rate in Figure 4a)
- I was glad the comparison in Table 2 was included that showed the results of training on either randomly sampled real data or a curated subset to address the limitations that had been discussed so far in the paper. It is always a tricky trade-off trying to address rare failure modes directly since by definition they only make up a small fraction of the overall validation setting.

Overall the paper is incredibly thorough, covering a wide variety of evaluation settings, and digging into ways to get more specific tangible insights into where these models do well and poorly. I appreciate the effort it takes to go beyond a simple comparison of overall accuracy on ImageNet.

**Weaknesses:**

- Much of the paper focuses on specific weaknesses that arise due to limitations of the image generator (image brightness, object scale, complex scenes). And the authors argue that these limitations lead to particularly pronounced failure modes in the model trained on synthetic data. I don't know that I agree that these failures are in fact specific to models trained on synthetic data. Looking at Figures 3 and 8, all of the error modes are very highly correlated between models trained on either real or synthetic images. Models trained on real images show the exact same weaknesses to "person blocking" and darker images and smaller objects. Given that this is such a fundamental part of the analysis it is odd to me that it really doesn't stand out as a synthetic data issue in particular, but instead as a more general way these models fail.

- One issue I have is that the authors do not discuss how much data goes into training the image generator. I feel like this is important context to the investigation. As far as I understand, Stable Diffusion was trained on LAION which consists of billions of image-text pairs. And my mental model of these image generators is that it is not unreasonable for them to reproduce source images almost perfectly. So it's not like this is truly "synthetic", some fraction of the data being generated probably bears high similarity to real source images with corresponding captions that match ImageNet labels. If anything, it is surprising that a model trained on that much data still does not know the difference between a rooster and a hen. So do the billions of images used to train StableDiffusion not count when discussing how much data is needed to get these results?

- I find it interesting that some of the specific failure modes seem easily addressed by data augmentation. There is focus in Section 3 on how to prompt the image generator to produce smaller/larger objects or whether other generators don't have the same light/dark limitations, but cropping and resizing and brightness augmentations could all trivially expose the model to more diverse object sizes and image conditions, no?

- The citation provided discussing whether real world data is limited (Villalobos et al 2024) makes a comment about there only being a couple trillion images taken each year while in this investigation we are looking at the impact of tens of thousands of images. How much are we at risk of running out of data?

**Questions:**

While maybe somewhat orthogonal, were any tests of data augmentations done to address any of the effects discussed here? Seems like data augmentation could complement some of what is more difficult to control in the image generation process.

It is interesting that we can train classifiers directly on the output of generative models. Given that, the real question someone might want answered is how to get a strong classifier with large numbers of unlabeled/weakly-labeled images and a small amount of real labeled data. I am not sure the most convincing case is being made that is worth it go through the roundabout process of producing millions of outputs from a generative model while mixing in a little bit of real data. How does performance compare to sampling and training on say, a million LAION images whose captions match up to the imagenet labels? Would that be a reasonable point of comparison? Similarly how would this compare to a model that is pretrained directly on LAION in the style of CLIP?

On a related note, it might helpful context to report how other pretrained models fare when finetuned on a small random subset of ImageNet. I am curious how other modern self-supervised pretraining strategies might do for example.

---

> ### Author Response · Authors · 2024-11-22
> **Response to Reviewer URhH**
>
> Thanks for your detailed reading and appreciating the exposition and evaluation of our work. We have revised the paper carefully following your suggestions. Below, we address your main concerns on the paper content.
>
> ---
>
> Q1: I don't know that I agree that these failures are in fact specific to models trained on synthetic data. Looking at Figures 3 and 8, all of the error modes are very highly correlated between models trained on either real or synthetic images.
>
> A1: Indeed, the dataset itself (whether real or synthetic data) may inherently contain these failure modes. However, in our paper, **we aim to compare the additional errors present in synthetic classifiers compared to real classifiers.** Therefore, under the condition where other influencing factors are held constant (i.e., model architecture, training process), we compared two settings: (1) equal data size and (2) equal ImageNet accuracy. The results show that, **even with equal accuracy, synthetic classifiers still perform worse in certain rare scenarios compared to real classifiers.** For instance, CLIP-Syn-371M exhibited a 10.84% lower performance on larger objects, 5.38% lower on darker images, and 13.65% lower on brighter images compared to CLIP-Real-64M, as illustrated in Figure 3. **This indicates that these errors stem from deficiencies in the synthetic data compared to real data.**
>
> Additionally, we apologize for the mistake in the y-axis scale of Figure 3 and Figure 8. The y-axis scale should have been **enlarged by a factor of 100**. This has been corrected in the revised version. From the modified Fig. 3, it is evident that synthetic classifiers perform significantly worse than real classifiers in certain rare scenarios.
>
> ---
>
> Q2：One issue I have is that the authors do not discuss how much data goes into training the image generator. … So do the billions of images used to train StableDiffusion not count when discussing how much data is needed to get these results?
>
> A2: This is a very good question! The amount of data used to train a generative model is indeed an important additional dimension. In the comparison and analysis, we only consider the amount of data required to train classifiers, that is, **we only consider the amount of synthetic data used to train the synthetic classifier followed [1], which ensures a fair comparison with the amount of data required to train real classifiers.** Since Stable Diffusion is a public open source model, training Stable Diffusion is not our cost when training synthetic classifeirs. Additionally, **if we were to consider the data used to train the generative model, it would imply that synthetic classifiers, which consumes a large amount of data, still performs worse than real classifiers.** This further validates the shortcomings of synthetic data in model training.
>
> **Reference:**
>
> [1] Lijie Fan, Kaifeng Chen, Dilip Krishnan, Dina Katabi, Phillip Isola, and Yonglong Tian. Scaling laws of synthetic images for model training... for now. In CVPR, 2024.
>
> ---
>
> Q3: I find it interesting that some of the specific failure modes seem easily addressed by data augmentation. There is focus in Section 3 on how to prompt the image generator to produce smaller/larger objects or whether other generators don't have the same light/dark limitations, but cropping and resizing and brightness augmentations could all trivially expose the model to more diverse object sizes and image conditions, no?
>
> While maybe somewhat orthogonal, were any tests of data augmentations done to address any of the effects discussed here?
>
> A3: Whether the failure modes can be solved through augmentation is a question that is worth studying. We conducted a more detailed ablation study on augmentation. **We compare models without augmentation (vanilla), brightness adjustment augmentation, blocking augmentation, crop augmentation, and models with a combination of brightness adjustment, blocking, and cropping.** The results are as follows. The performance gap between models with and without augmentation is minimal, and the difference between SynTune and RealTune after using augmentation has not decreased. This indicates that augmentation cannot easily addressed synthetic data failure modes. **What matters is RealTune**, underscoring the importance of real data.
>
> The results are added to **Appendix B**.
>
> |  | Vanilla | Bright | Block | Crop | Bright+Block+Crop |
> | --- | --- | --- | --- | --- | --- |
> | SynTune | 50.81 | 51.03 | 50.59 | 51.47 | 51.81 |
> | RealTune | 58.86 | 59.03 | 59.15 | 59.96 | 59.95 |
>
> *CLIP-Syn-371M ImageNet accuracy under different augmentation methods*
>
> |  | Vanilla | Bright | Block | Crop | Bright+Block+Crop |
> | --- | --- | --- | --- | --- | --- |
> | SynTune | 58 | 58.60 | 57.26 | 57.41 | 55.48 |
> | RealTune | 70.51 | 71.21 | 71.21 | 71.17 | 69.98 |

---

> ### Author Response · Authors · 2024-11-22
> **Response to Reviewer URhH**
>
> Q4: The citation provided discussing whether real world data is limited (Villalobos et al 2024) makes a comment about there only being a couple trillion images taken each year while in this investigation we are looking at the impact of tens of thousands of images. How much are we at risk of running out of data?
>
> A4: The issue of running out of data is indeed a controversial topic. From our perspective, although humans generate vast amounts of real data, **there is not much high-quality data.** In contrast, generative models, with the ability to produce data in specified directions under our control, may possess higher quality. This is why we compare the differences between real and synthetic data in model training.
>
> **Since the process of generating data is time-consuming and resource-intensive, we conducted the experiment in a more controllable setting.** This is why our experiments’ scale do not reach the trillion-level. To ensure fair comparisons, we examined two settings: (1) equal data size and (2) equal accuracy, as quantitative experiments on real-world scenarios. **The results indicate that due to data quality issues, there are still discrepancies when using synthetic data for model training. However, these discrepancies can be quickly bridged with a small amount of real data.** Therefore, designing appropriate training strategies can enhance the effectiveness of synthetic data.
>
> ---
> Q5:  The real question someone might want answered is how to get a strong classifier with large numbers of unlabeled/weakly-labeled images and a small amount of real labeled data. I am not sure the most convincing case is being made that is worth it go through the roundabout process of producing millions of outputs from a generative model while mixing in a little bit of real data. How does performance compare to sampling and training on say, a million LAION images whose captions match up to the imagenet labels? Would that be a reasonable point of comparison? Similarly how would this compare to a model that is pretrained directly on LAION in the style of CLIP?
>
> A5: There are numerous works currently exploring representation learning using synthetic data [1,2,3,4,5]. [1] demonstrates that the accuracy of CLIP trained on synthetic data generated by Stable Diffusion can surpass CLIP trained on real data. [2] shows that when the generative model does not use additional real data, the model trained by mixing synthetic data and real data can also outperform the model trained on real model. **What we want to study in our paper is whether synthetic classifiers still have some common defects even if the synthetic model reaches the accuracy of the real model, and whether we can further unleash its potential by identifying and solving the existing problems.** The results show that synthetic classifiers do have defects and these defects can be quickly remedied by finetuning a small amount of real data.
>
> **Reference:**
>
> [1] Lijie Fan, Kaifeng Chen, Dilip Krishnan, Dina Katabi, Phillip Isola, and Yonglong Tian. Scaling laws of synthetic images for model training... for now. In CVPR, 2024.
>
> [2] Yifei Wang, Jizhe Zhang, and Yisen Wang. Do generated data always help contrastive learning? In ICLR, 2024.
>
> [3] Shekoofeh Azizi, Simon Kornblith, Chitwan Saharia, Mohammad Norouzi, and David J Fleet. Synthetic data from diffusion models improves imagenet classification. Transactions on Machine
> Learning Research, 2023.
>
> [4] Yonglong Tian, Lijie Fan, Phillip Isola, Huiwen Chang, and Dilip Krishnan. Stablerep: Synthetic images from text-to-image models make strong visual representation learners. In NeurIPS, 2023.
>
> [5] Alceu Bissoto, Eduardo Valle, and Sandra Avila. Gan-based data augmentation and anonymization for skin-lesion analysis: A critical review. In CVPR, 2021.

---

> ### Author Response · Authors · 2024-11-22
> **Response to Reviewer URhH**
>
> Q6: On a related note, it might helpful context to report how other pretrained models fare when finetuned on a small random subset of ImageNet. I am curious how other modern self-supervised pretraining strategies might do for example.
>
> A6: Thank you for your insightful recommendation! Due to the extensive time requirements of conducting experiments on ImageNet, we conduct **experiments on DINO using ImageNet100.** The random subset used for RealTune consists of randomly selecting 100 images per class in ImageNet100 (~7.7% of ImageNet100). The results are as follows. The results indicate that **without RealTune, DINO-Syn exhibits an accuracy 32.72% lower than DINO-Real**, which is a significant gap. However, **after RealTune was applied to DINO-Syn, the performance gap between DINO-Syn and DINO-Real narrows to 7.62%** (DINO-Real Vanilla v.s. DINO-Syn RealTune acc), further highlighting the effectiveness of RealTune in self-supervised learning. The results are added to **Appendix B**.
>
> *ImageNet100 classification accuracy of different finetune methods on DINO. DINO-Real represents pretraining on Real ImageNet100, while DINO-Syn represents pretraining on synthetic ImageNet100.*
>
> |  | Vanilla | SynTune | RealTune |
> | --- | --- | --- | --- |
> | DINO-Syn | 35.48 | 32.96 | 60.58 |
> | DINO-Real | 68.2 | 39.28 | 63.54 |
>
> ---
>
> Thank you again for your careful reading. We have carefully refined our paper following your suggestions, and addressed each of your concerns above. We respectfully suggest that you could re-evaluate our work based on the updated results. We are very happy to address your remaining concerns on our work.

---

> > ### Author Response · Authors · 2024-11-25
> >
> > Dear Reviewer URhH,
> >
> > Thank you for appreciating our effort for analyzing beyond a simple comparison of overall accuracy on ImageNet.
> >
> > We have carefully prepared a detailed response to address each of your questions. Would you please take a look and let us know whether you find it satisfactory?
> >
> > We note that Reviewer rppm has appreciated our response and raised the score beyond the acceptance bar. We respectfully suggest that you could re-evaluate our work with the updated explanations and results.
> >
> > If you have any other questions, please feel free to ask.
> >
> > Thanks! Have a great day!
> >
> > Authors

---

> > > ### Comment · Reviewer_URhH · 2024-11-27
> > >
> > > First off, I want to say the authors have been incredibly thorough in their response, and I know how tough it is getting little to no response from the reviewers.
> > >
> > > I have a slightly different interpretation of the data augmentation results. The key focus of this paper is that synthetic data on its own is not particularly good and we can introduce a little bit of real data to bridge the gap. This is great. A large analysis follows into why synthetic data falls short and the authors direct our attention to a few particular types of error modes. Turns out adding data augmentation to target those error modes doesn’t bridge the gap the same way that real data does. This suggests to me that those error modes are perhaps not worth the attention they are getting in this paper for why synthetic data is not yielding better performance. It is likely that even if the synthetic data could capture light/dark scenes and varying object scales we would see a big gap in performance without introducing real data.
> > >
> > > I would be curious what fraction of overall accuracy is affected by the rare scenarios, I suspect very little. The improvement in performance with RealTune is significant. It feels as though it must be targeting some other way that synthetic data is insufficient. Perhaps something we would not be able to visibly perceive ourselves that distinguishes synthetic images from real images (e.g. some sort of high frequency image statistics). For example, if we look at the improvements made in Table 2, the last two rows suggest to me that we are probably seeing the effect of a more general improvement to model behavior rather than having real data fill in these specific missing error modes.
> > >
> > > I do think overall the point being made about fine-grained categories is stronger. It does seem likely that the inability of the generative model to distinguish and appropriately produce samples for such categories plays a role here.
> > >
> > > Ultimately, I don’t know that this paper has yet offered a clear picture as to what specifically the small fraction of real images brings to the table to improve performance. This analysis can make for a strong paper, but at the moment it still falls short to me. And it is worth saying, this sort of analysis is difficult, overall this is an impressive and challenging undertaking by the authors, and an important direction!
> > >
> > > Minor points:
> > > - it would be interesting to see the imagenetX results after data augmentation
> > > - any thoughts for why DINO-Real + RealTune leads to a drop in performance?

---

### Official Review · Reviewer_YzjF · 2024-11-09

**Soundness:** 2
**Presentation:** 2
**Contribution:** 2
**Rating:** 3
**Confidence:** 3

**Summary:**

The paper presents a quantitative study of closed-set classification accuracy of models trained on synthetic vs real data across various settings, vanilla ImageNet testing, finegrained classification over ImageNet finegrained classes and a list of rare scenarios defined. The paper also ablates the impact of classifier-free guidance scale parameter, text prompt design, generative model over the trained model accuracy using thus generated synthetic data. It also identifies three specific factors that lead to poorer accuracy for models trained purely with synthetic data on object scale, brightness and occlusion by human. Then the paper proposes to fine-tune using a small amount of real data to mitigate the lower accuracy of model trained with synthetic data on these scenarios.

**Strengths:**

The paper covers a number of quantitative tests of comparisons of models trained with real vs synthetic data. The motivation to go deep into the training data generated by SD to understand where potentially the gap of downstream trained classifier could come from makes sense.

**Weaknesses:**

The described scenarios are rather limited and narrow and lack of generalization to broader concepts, e.g., scale, occlusion with human are very specific settings.

Lack of comprehensive evaluation. There is no comparison of baselines with simple data augmentation on object scale and pixel occlusion. Also, Figure 4b mentions class imbalance as an issue for downstream model training. For this specific case, do class rebalancing techniques help solve the issue already? similarly for scale and occlusions, there is also no comparisons with baselines with simple data augmentation techniques.

In Figure2, the gap between similar ImageNet accuracy models trained with real and synthetic data over the finegrained domain does not appear to be very large - for example. 83.5 of CLIP-REAL-64M vs 83.1 of CLIP-Sync-371M, and 86.21 of ViT-Real-0.25M vs 85.4 of ViT-Sync-2M. It is unclear if these accuracy gaps are statistically significant or within the regime of model sensitivity. Thus it is unclear if the claim of synthetically trained models performing worse on fine-grained categories is technically sound or not.

Lack of technical novelty. The idea of pretraining with synthetic data and then fine-tune using a small amount of real data has been adopted in many prior works [1] [2].

[1] Task2Sim: Towards Effective Pre-training and Transfer from Synthetic Data
[2] From Fake to Real: Pretraining on Balanced Synthetic Images to Prevent Bias

**Questions:**

In Figure6, do the vanilla and SynTune baseline also use the additional 40K real samples used in the proposed RealTune setting? if not, it's not surprising that RealTune gets better since it leverages additional source of information.

It seems that here are cases where RealTune is worse than Vanilla, e.g., ViT-Real-1M setting, why is that?

---

> ### Author Response · Authors · 2024-11-22
> **Response to Reviewer YzjF**
>
> We appreciate your careful reading and pointing out the problems. We have diligently revised our paper in accordance with your suggestions. Below we address your main concerns.
>
> ---
>
> Q1: The described scenarios are rather limited and narrow and lack of generalization to broader concepts, e.g., scale, occlusion with human are very specific settings.
>
> A1: We followed the ImageNet-X[1] to identify common defects in synthetic vision models in real-world scenarios. These specific typical issues reflect gap between synthetic data and real data in model training. **This analysis can be extended to other domains such as text tasks.**
>
> We compared the performance of GPT2 models trained using an equal amount (0.11 million) of real texts and synthetic texts generated by GPT4. The results are presented in the table below. The results indicate that similar to the results in vision tasks, there is a significant performance gap between GPT2-Syn and GPT2-Real, further confirming the limitations of synthetic data in model training. Additionally, we noted that RealTune led to a 0.7 decrease in loss for GPT2-Syn, reducing the disparity with GPT2-Real, while SynTune resulted in a 0.22 increase in loss for GPT2-Syn. This highlights the effectiveness of RealTune for text tasks as well.
>
> Experimental results are added in **Appendix B**.
>
> *GPT2 loss of different finetune methods. GPT2-Real represents a model pretrained on real data, and GPT2-Syn represents a model pretrained on an equivalent amount of synthetic data.*
>
> |  | Vanilla (no finetune) | SynTune | RealTune |
> | --- | --- | --- | --- |
> | GPT2-Real | 2.78 | 3.21 | 3.04 |
> | GPT2-Syn | 4.29 | 4.51 | 3.59 |
>
> **References:**
>
> [1] Badr Youbi Idrissi, Diane Bouchacourt, Randall Balestriero, Ivan Evtimov, Caner Hazirbas, Nicolas
> Ballas, Pascal Vincent, Michal Drozdzal, David Lopez-Paz, and Mark Ibrahim. Imagenet-x: Understanding model mistakes with factor of variation annotations. In ICLR, 2023.
>
> ---
>
> Q2: Lack of comprehensive evaluation. (1) There is no comparison of baselines with simple data augmentation on object scale and pixel occlusion.  (2) Also, Figure 4b mentions class imbalance as an issue for downstream model training. For this specific case, do class rebalancing techniques help solve the issue already
>
> A2：Thank you for your suggestions.
>
> > There is no comparison of baselines with simple data augmentation on object scale and pixel occlusion.
> >
>
> **(1)** To further investigate the impact of augmentation, we conducted a  detailed ablation study. **We compared models without augmentation (vanilla), brightness adjustment augmentation, blocking augmentation, crop augmentation, and models with a combination of brightness adjustment, blocking, and cropping.** The results are as follows. The performance gap between models with and without augmentation is minimal, and the gap between SynTune and RealTune after using augmentation does not decreased. This indicates that augmentation does not address the shortcomings of synthetic classifiers and does not play a significant role here. What matters is RealTune, underscoring the importance of real data. The results are added to **Appendix B**.
>
>
> *ViT-Syn-1M ImageNet accuracy under different augmentation methods*
>
> |  | Vanilla | Bright | Block | Crop (ours) | Bright+Block+Crop |
> | --- | --- | --- | --- | --- | --- |
> | SynTune | 50.81 | 51.03 | 50.59 | 51.47 | 51.81 |
> | RealTune | 58.86 | 59.03 | 59.15 | 59.96 | 59.95 |
>
> *CLIP-Syn-371M ImageNet accuracy under different augmentation methods*
>
> |  | Vanilla | Bright | Block | Crop (ours) | Bright+Block+Crop |
> | --- | --- | --- | --- | --- | --- |
> | SynTune | 58 | 58.60 | 57.26 | 57.41 | 55.48 |
> | RealTune | 70.51 | 71.21 | 71.21 | 71.17 | 69.98 |
>
> > Also, Figure 4b mentions class imbalance as an issue for downstream model training. For this specific case, do class rebalancing techniques help solve the issue already
> >
>
> **(2)** This seems a misunderstanding of our Figure 4b. We are not suggesting that the synthetic dataset itself exhibits imbalance issues. Instead, we aim to **deliver that due to generative models' tendency to confuse fine-grained classes during generation, it leads to imbalanced predicted labels.** Essentially, **we are still emphasizing that generative models are prone to confusing fine-grained classes.** To make it more easily understandable, we **replace Figure 4(b) from a predicted label histogram to a fine-grained confusion matrix of 8 snake species for a more intuitive understanding of fine-grained class confusion** in the revision.

---

> ### Author Response · Authors · 2024-11-22
> **Response to Reviewer YzjF**
>
> Q3: In Figure2, the gap between similar ImageNet accuracy models trained with real and synthetic data over the finegrained domain does not appear to be very large - for example. 83.5 of CLIP-REAL-64M vs 83.1 of CLIP-Sync-371M, and 86.21 of ViT-Real-0.25M vs 85.4 of ViT-Sync-2M. It is unclear if these accuracy gaps are statistically significant or within the regime of model sensitivity. Thus it is unclear if the claim of synthetically trained models performing worse on fine-grained categories is technically sound or not.
>
> A3: The ViT and CLIP checkpoints we used from paper[1]. And paper[1] did not report the models' performance variation or provide multiple checkpoints for us to calculate it. And it was challenging for us to re-pretrain ViT and CLIP due to the computational demands of generating synthetic data and pretraining. However, it is generally believed that the variation at the ImageNet level is within 0.3. This is a common practice in ImageNet-scale experiments, and in real world, the variance/stdev is usually much smaller. For example, in ViT paper [2], they report 85.30 ± 0.02 for ViT and 87.54 ± 0.02 for ResNet (Table 2 [2]). Therefore, we believe that a difference by 0.4 (83.5 vs. 83.1) can be considered as a clear difference on ImageNet. Though, we do acknowledge that it is not a very large gap, and the gap is clearer when comparing equal data sizes. We have added this to the footnote to avoid misunderstanding.
>
> ---
>
> Q4: Lack of technical novelty. The idea of pretraining with synthetic data and then fine-tune using a small amount of real data has been adopted in many prior works [1] [2].
>
> A4: Using  synthetic data for pretraining and real data for finetuning has indeed been explored in many studies. However, the design for using both is not the primary focus of our paper. **What we aim to investigate is whether real data can bridge the significant gap between synthetic and real classifiers, and at what cost this gap can be bridged.** Our research in Section 4 reveals that by **using a very small amount of real data for a short finetuning period**, we can rapidly bridge this gap, a finding that has not been revealed in previous articles.
>
> For papers [1,2], we conduct a detailed comparison of the differences between our method and theirs as follows:
>
> **Paper [1].** We note that although they utilize synthetic data to pretrain a model and real data for fine-tuning the model, [1] primarily focuses on using real data and a network (Task2Sim) to select hyperparameters for generating synthetic images most beneficial for downstream tasks. Unlike them, **we do not need to modify the hyperparameters of the generative model.** Additionally, [1] uses the entire dataset for finetuning, whereas **we only require a random subset of a dataset (e.g., 3% of ImageNet).** These aspects highlight the efficiency and effectiveness of our approach.
>
> **Paper [2].** This is a work closely related to ours. However, we find that in Appendix E of their paper, it is mentioned that "All models are also pretrained on ImageNet." This indicates that **their method actually involves three stages: (1) pretraining on ImageNet, (2) pretraining on synthetic data, and (3) finetuning on real data.** So this approach is much more **cumbersome and time-consuming** compared to ours. Additionally, **they also use the entire real dataset for finetuning.** These comparisons highlight the effectiveness and simplicity of our approach.
>
> > [1] Task2Sim: Towards Effective Pre-training and Transfer from Synthetic Data
> >
>
> > [2] From Fake to Real: Pretraining on Balanced Synthetic Images to Prevent Bias
> >

---

> > ### Author Response · Authors · 2024-11-22
> > **Response to Reviewer YzjF**
> >
> > Q5: In Figure 6, do the vanilla and SynTune baseline also use the additional 40K real samples used in the proposed RealTune setting? if not, it's not surprising that RealTune gets better since it leverages additional source of information.
> >
> > A5: **Vanilla and SynTune do not use additional real samples, but SynTune uses an additional 40k synthetic samples for fine-tuning.** Here, **we explore the impact of the same quantity but different sources of data on fine-tuning through a comparison between RealTune and SynTune.** The results in Figure 6 indicate that RealTune significantly enhances the model (Vanilla) performance, while SynTune either harms the model (Vanilla) performance or shows insignificant improvement. This suggests that data quality is crucial for finetuning, highlighting the value of RealTune.
> >
> > ---
> >
> > Q6: It seems that here are cases where RealTune is worse than Vanilla, e.g., ViT-Real-1M setting, why is that?
> >
> > A6: Indeed, ViT-Real-1M performs worse after RealTune. This is essentially because ViT-Real-1M is already pretrained on 1M real data, and finetuning on a small 40k real data do not bring extra information but potentially leads to overfitting or forgetting, resulting in the lower performance. This shows that the 40k real data itself do not provide much extra information but they are sufficient to boost synthetic classifiers a lot, showing the significance of RealTune.
> >
> > ---
> >
> > Once more, we extend our gratitude for your careful review. We have carefully refined our paper according to your suggestions. We kindly invite you to re-examine our work in light of these updated improvements. We are pleased to resolve any further issues or concerns you may have regarding our work.

---

> > > ### Author Response · Authors · 2024-11-25
> > >
> > > Dear Reviewer YzjF,
> > >
> > > We have carefully prepared a detailed response to address each of your questions. Would you please take a look and let us know whether you find it satisfactory?
> > >
> > > We note that Reviewer rppm has appreciated our response and raised the score beyond the acceptance bar. We respectfully suggest that you could re-evaluate our work with the updated explanations and results.
> > >
> > > If you have any other questions, please feel free to ask.
> > >
> > > Thanks! Have a great day!
> > >
> > > Authors

---

### Author Response · Authors · 2024-11-22
**Paper Update**

We sincerely thank all reviewers for their thorough reading and valuable feedback. We have thoughtfully addressed their concerns and integrated these suggestions into the revised manuscript. The key revisions include:

- **The ImageNet-X evaluation results are scaled up 100 times (Figure 3, Figure 8, Table 2, Table 5).** We apologize for forgetting to multiply by 100% at first. From the modified results, it is evident that synthetic classifiers perform significantly worse than real classifiers in certain rare scenarios.
- Sec 2.1 Added discussion of the model’s evaluation results on ImageNet-C and ImageNet-3DCC from paper [1].
- Sec 3.1 replaced Figure 4(b) from a predicted label histogram to a fine-grained confusion matrix of 8 snake species.
- Sec 4：Added analysis of the results in Table 3 in section4.3; Added RealTune results on GPT2 in section 4.4.
- Sec 5: A part of the related works are moved to Appendix C due to space constraints.
- **Appendix B: add many new analysis results** on 1) comparison of different mixing methods, 2) results on different ratio of real and synthetic data, 3) ablation study of data augmentation, 4) DINO results 5) results on Caltech101 and EuroSAT 6) results on unbalanced datasets.

> [1] Krishnakant Singh, Thanush Navaratnam, Jannik Holmer, Simone Schaub-Meyer, and Stefan Roth. Is synthetic data all we need? benchmarking the robustness of models trained with synthetic images. In CVPR, 2024.

---

### Meta-Review · Area_Chair_mG7b · 2024-12-19

**Metareview:**

This paper investigates the classifiers trained on synthetic vs real datasets using CLIP and ViT, on different benchmarks such as fine-grained classification. The results show that synthetic classifiers exhibit deficiencies, and with the proposed RealTune approach, the classifier performance can be significantly improved with a small set of real data. The paper provides a number of evaluations, and the presentation is of high quality. However, the novelty of the work is still limited given the focus of the work is narrow, and the effectiveness of RealTune is also limited in some evaluations. I would recommend rejecting this work.

**Additional Comments On Reviewer Discussion:**

Reviewer YzjF asked for more comprehensive evaluations and a more rigorous analysis of ImagetNet accuracy. The performance of RealTune is also sometimes worse than ViT-Real-1M. The authors provided more evaluations on GPT2 and gave reasons why RealTune can be worse. The rebuttal is very detailed, but judging from the result on ImageNet, the improvement is still considered very small.

Reviewer URhH questioned whether the mage generators are really the cause, and also raised points regarding data augmentation. The authors gave answers according to the specific result in the paper. It seems the reviewer still did not quite agree with the reasons regarding the data augmentation provided by the authors.

Reviewers 5XMN and YzjF are both concerned about the novelty of the work. The authors restated that the work focuses on the evaluation of synthetic classifiers, not using synthetic data to train models. For the final decision, this is one of the main factors for rejection.

Reviewer rppm asked about the generalization of the approach to other domains and the importance of the quality of fine-tuning data. The authors resolved the concerns with extra experiments and data points.

---

### Decision · Program_Chairs · 2025-01-22

Reject